# *Code Researcher*: DEEP RESEARCH AGENT FOR LARGE SYSTEMS CODE AND COMMIT HISTORY

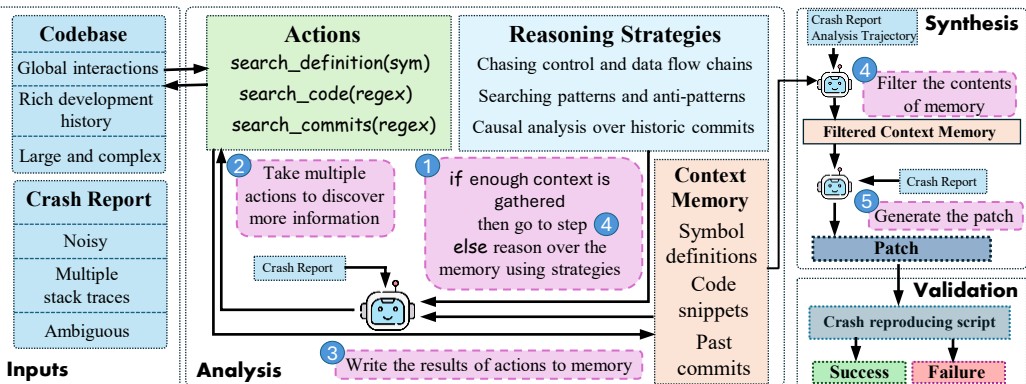

Figure 1: *Code Researcher* conducts deep research over code in three phases: (1) Starting with the codebase and crash report as input, the ANALYSIS phase performs multi-step reasoning about semantics, patterns, and commit history of code. It gathers context in a memory. (2) The SYNTHESIS phase filters the contents of the memory to keep relevant context and generates a patch. (3) The VALIDATION phase uses external tools to validate the patch.

## ABSTRACT

Large Language Model (LLM)-based coding agents have shown promising results on coding benchmarks, but their effectiveness on systems code remains underexplored. Due to the size and complexities of systems code, making changes to a systems codebase requires *researching* about many pieces of context, derived from the large codebase and its massive commit history, *before* making changes. Inspired by the recent progress on deep research agents, we design the first deep research agent for code, called *Code Researcher*, and apply it to the problem of generating patches to mitigate crashes reported in systems code. *Code Researcher* performs multi-step reasoning about semantics, patterns, and commit history of code to retrieve all relevant context from the codebase and its commit history. We evaluate *Code Researcher* on kBenchSyz (Mathai et al., 2024), a benchmark of Linux kernel crashes, and show that it significantly outperforms strong baselines, achieving a crash-resolution rate (CRR) of 48%, compared to 31.5% by SWE-agent (Yang et al., 2024) and 31% by Agentless (Xia et al., 2024), using OpenAI's GPT-4o model. Scaling up sampling budget to 10 trajectories increases *Code Researcher*'s CRR to 54%. *Code Researcher* is also robust to model choices, reaching 67% with the newer Gemini 2.5-Flash model. Through another experiment on an open-source multimedia software, we show the generalizability of *Code Researcher* and also conduct ablations. Our experiments highlight the importance of global context gathering and multi-faceted reasoning for large codebases.

## 1 INTRODUCTION

Automating coding using Large Language Models (LLMs) and LLM-based agents is a very active area of research. Popular benchmarks like LiveCodeBench (Jain et al., 2024) and SWE-bench (Jimenez et al., 2024) respectively test coding abilities on standalone competitive coding problems and GitHub issues over library or application code. Despite the demonstrated progress of coding agents on these benchmarks, they are yet to scale to complex tasks over an important class of code, *systems code*.

Systems code powers critical and foundational software like operating systems, networking stacks, cloud infrastructure and system utilities. Systems codebases have multiple dimensions of complexity. Firstly, they are *very large*, with thousands of files and millions of lines of code. Secondly, systems code often interfaces directly with the hardware and is performance critical. This results in *complex low-level code* (involving pointer manipulations, compile-time macros, etc.) in languages like C/C++, and *global interactions* between different parts of the codebase for concurrency, memory management, maintenance of data-structure invariants, etc. Finally, foundational systems codebases have *rich development histories* spanning years or even decades, containing contributions by thousands of developers, which are important references on legacy design decisions and code changes.

We consider the problem of generating patches to mitigate crashes reported in systems code. This is a uniquely challenging setting, as opposed to the SWE-bench (Jimenez et al., 2024) like setting tackled by many prior works (Yang et al., 2024; Jain et al., 2025; Ouyang et al., 2025) using LLMs and SLMs in coding agents. SWE-bench contains human-written issue descriptions from moderately-sized codebases, which explain the nature of the bug and might indicate which files are likely relevant. Coding agents (Yang et al., 2024; Wang et al., 2025) are designed to take advantage of this and quickly navigate the repository to reach the buggy files. In fact, an Agentless (Xia et al., 2024) approach, which has a simpler, fixed workflow of localizing the files to edit followed by repair, performs competitively on SWE-bench. These approaches do not expend much efforts in gathering codebase-wide, global context. In our setting, the bugs are described by stack traces which are devoid of natural language hints and contain a much larger number of files and functions than an issue description. Due to the nature of crash reports and the complex global interactions in large systems codebases, multi-step reasoning and context gathering become important.

Automating such complex tasks in systems codebases requires a different type of agents, agents that can *research* about many pieces of context, derived automatically from the large codebase and its massive commit history, *before* making changes. Recently, *deep research* agents have been developed to solve complex, knowledge-intensive problems that require careful context gathering and multi-step reasoning, before synthesizing the answer. The agents and techniques have mostly focused on long-form document generation or complex question-answering over web contents (Shao et al., 2024; OpenAI, 2025b; Google, 2025a; Perplexity, 2025; Li et al., 2025; Wu et al., 2025b) and enterprise data (Anthropic, 2025; Microsoft, 2025). Inspired by these advances, we propose the first deep research agent for code, called *Code Researcher*, and apply it to the problem of generating patches for mitigating crashes reported in systems code.

As shown in Figure 1, *Code Researcher* works in three phases: (1) ANALYSIS: Starting with the crash report and the codebase, this phase performs multi-step reasoning over semantics, patterns, and commit history of code. The "Reasoning Strategies" block shows the reasoning strategies used. Each reasoning step is followed by invocations of tools (labeled "Actions" in Figure 1) to gather context over the codebase and its commit history. The information gathered is stored in a context memory and when the agent is able to conclude that it has gathered sufficient context, it moves to the next phase. (2) SYNTHESIS: The SYNTHESIS phase uses the crash report, the context memory, and the reasoning trace of the ANALYSIS phase to filter out irrelevant memory contents. Then, it generates patches, which may edit one or more buggy code snippets from memory, possibly spread across multiple files. (3) VALIDATION: Finally, the VALIDATION phase checks if the generated patches prevent the crash from occurring using external tools. A successful patch is presented to the user.

We evaluate the effectiveness of *Code Researcher* on the kBenchSyz benchmark (Mathai et al., 2024), containing 279 Linux kernel crashes detected by the Syzkaller fuzzer (Google, 2025b). This benchmark is challenging because the Linux kernel (Torvalds, 1991) is a canonical example of a systems codebase with complex low-level code and massive size (75K files and 28M lines of code), and has rich development history. *Code Researcher* resolves $48\%$ of crashes using GPT-4o and 5 sampled patches, significantly outperforming the two strong and popular baselines of SWE-agent (Yang et al., 2024) ($31.5\%$) and Agentless (Xia et al., 2024) ($31\%$) in the same setting (and customized for the kernel crash resolution task). A concurrent work, CrashFixer (Mathai et al., 2025), explores a simpler setting where the agent is provided the ground-truth buggy files to edit (the *assisted* setting), whereas, *Code Researcher* takes only the crash report as input (the *unassisted* setting).

*Code Researcher* benefits from scaling inference compute, reaching $54\%$ with pass@10, and is robust to the choice of model, resolving $67\%$ of crashes with the newer Gemini 2.5-Flash LLM. It gathers context of high coverage and quality, exploring about 10 files per trajectory compared

to a much smaller number 1.33 of files explored by SWE-agent. We demonstrate the importance of causal analysis over historical commits, a novel feature of *Code Researcher*. To ensure that the proposed patches do not break existing functionality, we run Linux kernel unit tests on *Code Researcher*'s crash-resolving patches. Though expensive to run, this provides additional validation beyond the crash-reproduction testing in kBenchSyz. We give evidence of the generalizability of *Code Researcher* by experimenting on an open-source multimedia software, FFmpeg (FFmpeg, 2025), where it resolves $7/10$ crashes tested. We are making our implementation and all results available in the supplementary material for review. We plan to make these public upon paper publication.

In summary, we make the following main contributions:

**(1)** We design the first deep research agent for code, *Code Researcher*, capable of handling large systems code and resolving crashes. Recognizing the importance of commit history in systems code, we equip the agent with a tool to efficiently search over commit histories.

**(2)** We evaluate *Code Researcher* on the challenging kBenchSyz benchmark (Mathai et al., 2024) and achieve a crash resolution rate of $54\%$, outperforming strong baselines and showing robust performance across model choices, reaching $67\%$ with the newer Gemini 2.5-Flash model. We also demonstrate its generalizability through experiments on a multimedia software, FFmpeg.

**(3)** Through a comprehensive evaluation, we show (i) how our deep research agent outperforms agents that do not focus on gathering relevant context, (ii) that this advantage persists even if the existing SOTA agent is given higher inference-time compute, and (iii) that reasoning models improve performance significantly if given well-researched context. We thoroughly validate *Code Researcher*'s crash-resolving patches using the Linux kernel unit test-suite in addition to the validation setup in kBenchSyz, providing confidence that they do not break existing functionality. Further ablations show the importance of (i) causal analysis over historical commits and (ii) memory filtering.

## 2 RELATED WORK

The LLM-powered software development subfield has produced several coding agents (Yang et al., 2024; Xia et al., 2024; Wang et al., 2025; Zhang et al., 2024; Wadhwa et al., 2024), predominantly evaluated on SWE-bench (Jimenez et al., 2024). SWE-bench focuses on GitHub issues from small to medium-sized Python repositories. However, systems code, the focus of our work, presents unique challenges. We highlight and contrast key related work in this context. A related, but orthogonal line of exploration is long context reasoning. But it has its own challenges, as discussed in Appendix F.

**Coding agents** Agents like SWE-agent (Yang et al., 2024) or OpenHands (Wang et al., 2025) use a single ReAct-style (Yao et al., 2023) loop endowed with shell commands or specialized tools for file navigation and editing. However, they tend to explore a small number of files per bug, without gathering and reasoning over the relevant codebase-wide context. AutoCodeRover (Zhang et al., 2024) uses tools based on program structure to traverse the codebase (albeit limited to Python code). It performs explicit localization of the functions/classes to edit using these tools, and those are later repaired. *Code Researcher* does not explicitly localize the functions to edit; instead it gathers relevant context for patch generation and decides what to edit in the SYNTHESIS phase. Some recent coding agents construct a dependency graph of the repository (Ouyang et al., 2025; Chen et al., 2025), which they then explore using approaches like MCTS (Ma et al., 2025). However, such agents are (1) limited to Python code and (2) scale very poorly, making it impractical to use on codebases of the scale of the Linux kernel. *Code Researcher* instead uses simple and scalable tooling to handle such codebases easily. *Code Researcher* is also the first agent to use causal analysis over historical commits; this is critical to handling subtle bugs introduced by code evolution in long-lived systems codebases.

**Deep research agents** Deep research is a fast emerging subfield in agentic AI (Microsoft, 2025; OpenAI, 2025b; Google, 2025a; Perplexity, 2025), to tackle complex, knowledge-intensive tasks, that can take hours or days even for experts. Academic work so far has focussed on long-form document generation (Godbole et al., 2024; Shao et al., 2024), scientific literature review (Wu et al., 2025b; Gottweis et al., 2025), and complex question-answering (Li et al., 2025; Wu et al., 2025a) based on the web corpus. The key challenges in deep research for such complex tasks include (a) intent disambiguation, (b) exploring multiple solution paths (breadth of exploration), (c) deep exploration (iterative tool interactions and reasoning), and (d) grounding (ensuring that the claims in the response are properly attributed). Most of the aforementioned challenges also apply to our setting. To the best of our knowledge, our work is the first to design and evaluate a deep research strategy for complex

bug resolution in large codebases. Most recently, OpenAI's Deep Research model has been integrated with GitHub repos for report generation and QA over codebases (OpenAI, 2025a). However, (a) it does not support agentic tasks like bug fixing, and (b) their indexing technique does not scale to very large codebases like the Linux kernel, whereas we use scalable tools for search.

**Automated kernel bug detection and repair** Prior work for detecting Linux kernel bugs includes various types of sanitizers, e.g., Kernel Address Sanitizer (KASAN) (Google, 2025a), and the Syzkaller kernel fuzzer (Google, 2025b), an unsupervised coverage-guided fuzzer that tries to find inputs to crash the kernel. *Code Researcher*, complementary to this, generates patches from crash reports. We use some traditional software engineering concepts like deviant pattern detection (Engler et al., 2001) and reachability analysis (Nielson et al., 2015), but leverage LLMs to scale to large codebases. As noted earlier, CrashFixer (Mathai et al., 2025) targets Linux kernel crashes but assumes that buggy files are known *a priori*. This assumption is unrealistic for large codebases like the Linux kernel. In contrast, *Code Researcher* autonomously locates buggy files using general search tools.

## 3    DESIGN OF *Code Researcher*

Large systems codebases, owing to their critical nature, undergo strict code development and reviewing practices by expert developers. The bugs that still sneak in are subtle and involve violations of global invariants (e.g., a data structure should be accessed only after acquiring a lock) and coding conventions (e.g., use of a specific macro to allocate memory), and unintended side effects caused by past changes. To fix such bugs, an agent needs to gather sufficient context from the codebase and its commit history, before generating hypotheses about the cause of a bug and attempt to fix it. With this insight, we design our deep research agent, *Code Researcher*. As shown in Figure 1, *Code Researcher* comprises of three phases: (1) ANALYSIS, (2) SYNTHESIS and (3) VALIDATION. We present the key details of our design in this section and complement it with implementation details in Appendix B. We also explain an example trajectory of *Code Researcher* in Appendix C.

### 3.1    ANALYSIS PHASE

The ANALYSIS phase of *Code Researcher* is responsible for performing deep research to understand the cause of a reported crash. We equip this phase with (a) actions to efficiently search over the codebase and the commit history and (b) reasoning strategies for code. At each step, the actions taken so far along with their results are stored in a context memory, which is used to construct the prompt.

#### 3.1.1    ACTIONS TO SEARCH OVER CODEBASE AND COMMIT HISTORY

We support the following actions: (1) `search_definition(sym)`: To search for the definition(s) of the specified symbol, which can be the name of a function, struct, global constant, union or macro and so on. It can be optionally passed a file name to limit the search. (2) `search_code(regex)`: To search the codebase for matches to the specified regular expression. This is a simple yet powerful tool, which can be used for searching for any coding pattern such as call to a function, dereferences to a pointer, assignment to a variable and so on. (3) `search_commits(regex)`: To search for matches to a regular expression over commit messages and diffs associated with the commits. The regular expression offers expressiveness, e.g., to search for occurrence of a term ("memory leak") in the commit messages or coding patterns in code changes (diffs). In addition, the agent can invoke (4) `done` to indicate that it has finished the ANALYSIS phase and (5) `close_definition(sym)`: To remove the definition of a symbol from the memory if the symbol is deemed irrelevant to the task.

#### 3.1.2    REASONING STRATEGIES FOR CODE

We ask the agent to explore the codebase to figure out the root cause of a crash and gather sufficient context to propose a fix. We induce the following reasoning strategies through prompting to guide the exploration of the codebase and its commit history. As shown in Figure 1, each reasoning step is followed by one or more actions. Additionally, we present the agent with a simple scratchpad, where it can add important discoveries for future reference.

**Chasing control and data flow chains** The *control flow* (Nielson et al., 2015) of a code snippet refers to the functions that are called and the branches in it, including conditional statements, loops, `gotos`

and even conditional compilation macros. Given a crash report and some code, the agent is asked to reason about control flow to understand how execution flows between different functions and how it leads to the crash. Similarly, *data flow* (Nielson et al., 2015) refers to how the values of variables get passed to different functions and how one variable is used to define another. So the agent should also reason about how data flows in the code. As a result of this reasoning, the agent may invoke a `search_definition(sym)` action to search for the definition of `sym` if it suspects that `sym` may have something to do with the buggy behavior and needs more information about `sym` to confirm or dispel the suspicion. It can also use other actions as suitable, e.g., `search_code(x\s*=)` to look for assignments to a variable named `x`, with `\s*` indicating zero or more whitespaces.

**Searching for patterns and anti-patterns** Traditional software engineering literature thinks of bugs as anomalies – patterns of code that are deviant (Engler et al., 2001). It follows that, to diagnose and understand a bug, one can find frequent patterns in the repository and check if a given piece of code deviates from it. *Code Researcher* reasons about which behavior is common or "normal" and which code snippets look anomalous. It can perform a `search_code(regex)` action to search for these patterns and anti-patterns using regular expressions. A classic case is checking a pointer for null value after allocation. If the agent notices a missing null check for `ptr`, it can perform `search_code(if\s*\(ptr==NULL\))` to search for null checks throughout the codebase on `ptr`. Similarly, it can perform `search_code(ptr\s*=.*alloc\(.*\))` to search for all allocations to `ptr` to verify whether other parts of the codebase typically perform a null check or not.

**Causal analysis over historical commits** An interesting and challenging aspect of a codebase that has been in development for a long time, as many foundational systems codebases have, is the rich history of commits. Because of continuous development, it is likely that a new bug has some past commits that can prove helpful in understanding or solving it. Indeed, developers often reference other commits when they come up with patches. *Code Researcher* reasons about how the codebase has evolved and how that evolution is related to the crash report. It can issue a `search_commits(regex)` action to search over past commit messages and diffs. For instance, the regular expression `handle->size|crypto_fun\(` matches commits that add or remove a `handle->size` access, or a call to `crypto_fun`.

**Iterative process of deep research** As shown in Figure 1, in each reasoning step, *Code Researcher* is asked to decide if it has acquired sufficient context to understand and solve the crash. If yes, it moves to the next phase of synthesizing the patch (Section 3.2). Initially, the context is empty and it starts its reasoning process by analyzing the contents of the stack trace and the diagnostic information provided as input. At each step, the agent evaluates the context accrued so far and decides which lines of exploration to extend by issuing multiple search actions simultaneously.

## 3.2 SYNTHESIS AND VALIDATION PHASES

The contents of memory and the reasoning trace of the ANALYSIS phase are passed to the SYNTHESIS phase, along with the crash report. The ANALYSIS phase has the flexibility to follow multiple paths of inquiry simultaneously. It can thus end up collecting information that does not turn out to be relevant, which also happens when a human does research on some topic. In large codebases, this irrelevant information can be quite large and can overwhelm the prompt. Thus, the SYNTHESIS phase first filters the memory and discards (action, result) pairs that are deemed irrelevant to the task of fixing the crash. The agent then uses the filtered information to generate a hypothesis about the nature of the bug and a potential remedy, and the corresponding patch. Finally, in the VALIDATION phase, the patch is applied to the codebase, and the codebase is compiled. The reproducer program that had originally caused a crash is run. If the crash is reproduced, the patch is rejected. If not, it is accepted.

## 4 EXPERIMENTAL SETUP

**Benchmarks** We use a thoroughly validated, reproducible subset of **200 instances** from the kBenchSyz benchmark (Mathai et al., 2024) of 279 Linux kernel crashes found by the Syzkaller fuzzer (Google, 2025b). Each instance in the benchmark consists of (1) a reproducer file, containing the user-space program that triggers the crash, (2) the ground-truth commit that fixed the bug, and (3) the crash report at the parent commit of the fix commit (we run all tools at this parent commit). To show generalizability, we also evaluated *Code Researcher* on 10 recent crashes of an open-source

multimedia software, FFmpeg (FFmpeg, 2025). More details about the kBenchSyz benchmark and the FFmpeg dataset are in Appendix G and Appendix D respectively.

**Evaluation metrics** We compute Pass@k ($\mathbf{P}@k$) defined as $\mathbf{P}@k = 1$, if applying at least one of the $k$ candidate patches generated by the tool prevents the crash, or $\mathbf{P}@k = 0$ otherwise. We report **(1) C**rash **R**esolution **R**ate (**CRR**) which is average $\mathbf{P}@k$, **(2)** average **recall**, i.e., the fraction of files modified in the ground-truth commit (*the ground-truth buggy files*) in the set of files edited by the agent, averaged over the $k$ candidate patches, and **(3)** the percentage of candidate patches where **All**, **Any** or **None** of the ground-truth buggy files are edited. When a tool does not produce a patch (e.g., it runs out of LLM call budget), the set of edited files is assumed to be empty. All the metrics are averaged over the 200 instances in the benchmark.

**Baselines** We evaluate *Code Researcher* in the *unassisted* setting (i.e., the ground-truth buggy files that are part of the fix commits are not divulged to the tool) and compare it against the following baselines: **(1) o1** (OpenAI, 2024b) **and GPT-4o** (OpenAI, 2024a) **in the** *assisted* setting, i.e., we give the ground-truth files that are part of the fix commits and the crash report as input. We prompt the model to generate a hypothesis about the crash's cause and a patch. **(2) o1 and GPT-4o in the** *stack context* **setting**, where we give the contents of the files mentioned in the crash report as input besides the crash report. **(3) SWE-agent 1.0** (Yang et al., 2024), a SOTA coding agent on the SWE-bench benchmark, **in the** *unassisted* **setting**. For fairness, we add a Linux kernel-specific example trajectory and background about the kernel to its prompts. We sample $k$ (for Pass@$k$) SWE-agent trajectories independently using a temperature of 0.6. **(4) Agentless** (Xia et al., 2024), another top coding agent on the SWE-bench benchmark that uses a fixed workflow of localization followed by repair, **in the** *unassisted* **setting**. We add background about the kernel to its prompts and sample $k$ (for Pass@$k$) patches using the same LLM (GPT-4o) as *Code Researcher* and SWE-agent. **(5) CrashFixer** (Mathai et al., 2025), state-of-the-art agent for Linux kernel crash resolution, **in the** *assisted* **setting**, as it directly uses the ground-truth files to generate patches. If a patch fails to build or crashes with the ground-truth reproducer, it iteratively refines it using the respective error messages. We report their results with the Gemini 1.5-Pro-002 model (v. 2024-09-24) and more recent results with the Gemini 2.5-Pro model (v. 2025-06-17).

**Implementation, hyperparameters** We employ **GPT-4o** (v. 2024-08-06) for *Code Researcher* and for the competing tools. We also experiment with **o1** (v. 2024-12-17) in the SYNTHESIS phase of *Code Researcher*, and, to show that *Code Researcher*'s design is not tied to the OpenAI family of models, the newer **Gemini 2.5-Flash** (v. 2025-06-17) for both the ANALYSIS and SYNTHESIS phases. All our experiments have a context length limit of 50K tokens. In the ANALYSIS phase, we use a **temperature** of $0.6$ and independently sample $k$ trajectories. For the SYNTHESIS phase, we sample with increasing temperatures $(0, 0.3, 0.6)$ until the agent produces a correctly-formatted patch, with a maximum of 3 attempts. We allow all tools a budget of at most **max calls** LLM calls to generate a single patch. Please refer to Appendix G for details about the **crash reproduction setup**, the **prompts**, the **compute resources** and the **configurations** for *Code Researcher* and the baselines.

## 5 EXPERIMENTAL RESULTS

We evaluate *Code Researcher* across six dimensions. First, we compare its crash resolution ability against state-of-the-art coding agents and baselines. We then analyze the context gathering capabilities of different tools by studying (a) whether they are able to find the files that need to be edited, and (b) the coverage and quality of additional global context gathered. Then, we assess the impact of two design choices, historical commit analysis and context filtering, on *Code Researcher*'s performance. Finally, we further validate *Code Researcher*'s crash-resolving patches with unit tests and a qualitative analysis, and show its generalizability to another codebase.

### 5.1 RQ1: HOW EFFECTIVE ARE DIFFERENT TOOLS AT RESOLVING LINUX KERNEL CRASHES?

Our main results are presented in Table 1, organized by setting, namely, assisted, stack context, unassisted, and unassisted + test-time scaled (Section 4).

**1) The assisted setting baselines show strong performance, but under unrealistic assumptions.** Given the ground-truth buggy files, LLMs like GPT-4o (CRR of $36\%$) are quite capable of resolving crashes. A reasoning model like o1 is significantly better (CRR of $51\%$). Adding iterative feedback

Table 1: Crash resolution rate (CRR) for different tools on the kBenchSyz benchmark (200 bugs). LLMs used by the tools are in parentheses. *Results are from Mathai et al. (2025), out of 279 bugs.

| Setting | Tool | Max calls | P@k | CRR (%) |
|---|---|---|---|---|
| Assisted | GPT-4o | 1 | P@5 | 36.00 |
| | o1 | 1 | P@5 | 51.00 |
| | CrashFixer (Gemini 1.5 Pro-002)* | ≥ 4 | P@16 | 49.22* |
| | CrashFixer (Gemini 2.5 Pro) | ≥ 4 | P@16 | **70.00** |
| Stack context | GPT-4o | 1 | P@5 | 29.50 |
| | o1 | 1 | P@5 | **40.00** |
| Unassisted | Agentless (GPT-4o) | 4 | P@5 | 31.00 |
| | SWE-agent (GPT-4o) | 15 | P@5 | 31.50 |
| | *Code Researcher* (GPT-4o) | 15 | P@5 | 48.00 |
| | *Code Researcher* (GPT-4o + o1) | 15 | P@5 | 58.00 |
| | *Code Researcher* (Gemini 2.5-Flash) | 15 | P@5 | **67.00** |
| Unassisted + Scaled | SWE-agent (GPT-4o) | 30 | P@5 | 32.00 |
| | *Code Researcher* (GPT-4o) | 30 | P@5 | 47.50 |
| | SWE-agent (GPT-4o) | 15 | P@10 | 37.50 |
| | *Code Researcher* (GPT-4o) | 15 | P@10 | **54.00** |

from the compiler and the crash reproduction setup, CrashFixer achieves 49.22% CRR with an older Gemini model and 70% CRR with a newer Gemini reasoning model using P@16 with at least 4 max calls.

**2) The stack context setting reveals the difficulty of the practical unassisted setting.** The assumption that an oracle can tell us exactly which files need to be edited is impractical. The gap between the assisted (idealistic) setting and the practical unassisted setting is highlighted by the simple but effective stack context setting where models are given the contents of all the files mentioned in the crash report (truncated to the context length limit). This is a strong baseline because all the ground-truth buggy files are present in the crash report for 74.50% crashes in our dataset. o1 achieves a CRR of 40%, which is impressive, but 11% lower than its performance in the assisted setting.

**3) *Code Researcher* consistently outperforms baselines in the unassisted setting.** Fixing the LLM as GPT-4o, *Code Researcher* achieves a CRR of 48%, significantly outperforming the SWE-agent and Agentless baselines, both the stack context baselines, and even the assisted GPT-4o baseline. This indicates that the context gathered by *Code Researcher* is much more effective than giving file contents based on the crash report, or using multi-step hierarchical localization (as done by Agentless), and is even better than directly giving all the contents of the files to be edited. The context gathered by GPT-4o during ANALYSIS can be better utilized by the reasoning model o1 during SYNTHESIS, increasing *Code Researcher*'s CRR to 58%. This further increases to 67% using the newer (albeit not as advanced as Gemini 2.5-Pro) model, Gemini 2.5-Flash, for both phases, indicating that *Code Researcher*'s design is not tied to a specific LLM family.

**4) Increasing the number of trajectories helps, while making them longer does not.** We examine how scaling the total inference budget (max calls × num trajectories $k$) impacts the performance of *Code Researcher* and SWE-agent. Doubling the max calls budget, i.e., making the trajectories of the agents longer, has a negligible effect on the CRR, whereas increasing the number of trajectories sampled improves SWE-agent's CRR to 37.50% and *Code Researcher* (GPT-4o)'s CRR to 54.00%.

### 5.2 RQ2: DO THE TOOL-EDITED FILES MATCH THOSE MODIFIED IN DEVELOPER FIXES?

The information needed by a developer to fix a crash can be divided into (1) the code to be edited and (2) additional context. We study (1) here by examining whether tools edit the same files as developers, and study (2) in the next section. Since buggy files are already provided in the assisted setting, we focus on the stack context and unassisted settings. Full results appear in Table 2 (Appendix A).

***Code Researcher* achieves the highest recall across all baselines**, editing all ground-truth buggy files in nearly half of candidate patches and at least one in another ∼ 8%. "Recall" here measures

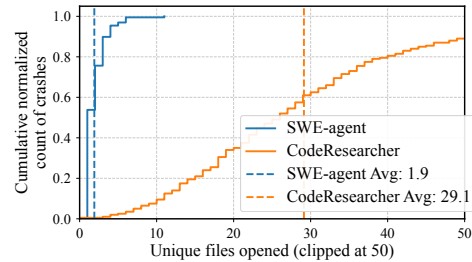

Figure 2: Files explored for each crash (summed over 5 trajectories).

the fraction of ground-truth files in those *edited*, not merely *explored*, making it notable that *Code Researcher* edits on average only 1.1 files while exploring 10 (Section 5.3). Agentless attains recall comparable to *Code Researcher* and much higher than SWE-agent, but resolves far fewer crashes. **This contrast underscores the importance of the global context gathered by *Code Researcher*, beyond just the code being edited.** Finally, scaling test-time compute (via increasing max calls or P@$k$) for *Code Researcher* and SWE-agent preserves the overlap between edited and ground-truth files, and using the newer Gemini 2.5-Flash for *Code Researcher* further improves its overlap.

### 5.3 RQ3: How effective is context gathering for resolving kernel crashes?

From Tables 1 and 2, *Code Researcher* (GPT-4o) significantly outperforms baselines that do not gather additional context but have high recall. This shows that gathering global context (instead of just localizing the code to edit) can drastically improve kernel crash resolution. Agentless, which does not gather any context, performs poorly, while SWE-agent does gather context from the codebase, yet it performs much worse than *Code Researcher*. Below, we investigate this further:

**1) Coverage of the context gathered:** Figure 2 shows the distribution of the number of unique files read across the 5 trajectories by *Code Researcher* and SWE-Agent (GPT-4o, P@5, 15 max calls). *Code Researcher* performs deep research over the codebase, reading 29.13 unique files across 5 top-level directories on average for each crash. In stark contrast, SWE-agent reads only 1.91 files on average for each crash. When averaged by trajectory, *Code Researcher* explores 10 unique files compared to only 1.33 files explored by SWE-agent.

**2) Overlap with developer-referenced context:** We use LLM-as-judge to determine the overlap of the context gathered by *Code Researcher* (GPT-4o) and SWE-agent with the context mentioned by the developer in the fix commit message (details in Appendix I). This context overlap is 54.18% (over candidate patches) for SWE-agent compared to 63.7% for *Code Researcher*, suggesting that *Code Researcher* does a much better job of identifying relevant context that the developer explicitly relied on when making the fix.

**3) Context quality when both edit all the ground-truth modified files:** To isolate the impact of the gathered context, we consider the subset of 90 crashes, where both *Code Researcher* and SWE-agent (using the same GPT-4o model) edit *all* the ground-truth files in *at least one candidate patch* generated by each tool. We can thus attribute their success (or failure) on this subset to the context gathered. *Code Researcher* resolves $55/90 = 61.10\%$ of crashes in this subset, while SWE-agent resolves only $34/90 = 37.78\%$ (discounting crash-resolving patches from each tool that do not edit *all* the ground-truth files). *Taken together, the three observations show that not only does Code Researcher gather more context than SWE-agent, it also gathers higher quality context.*

### 5.4 RQ4: How important are historical commit analysis and context filtering?

**Commit history analysis** *Code Researcher* is the first agent to explicitly leverage the rich development history of codebases. In this ablation, we run it without the search_commits action on the set of 96 bugs that were successfully resolved by *Code Researcher* (GPT-4o, Pass@5, 15 max calls). Table 3 (Appendix A) shows that removing the search_commits action leads to a 10% drop in the crash resolution rate, and decreases the ability to edit the ground-truth modified files. This highlights that the search_commits action plays a crucial role in context gathering

and localization. Notably, for the example in Appendix H, we also observe that *Code Researcher* navigates to the *same* buggy commit that originally introduced the bug being repaired.

**Context filtering** As mentioned in Section 3.2, the ANALYSIS phase often gathers large amounts of irrelevant context, especially in big codebases like the Linux kernel. The SYNTHESIS phase filters this memory before synthesizing a patch. We provide two pieces of evidence for the importance of this filtering. First, the average memory length (capped at $50K$) across all *Code Researcher* (GPT-4o, P@5, 15 max calls) trajectories dropped from $21,557$ tokens to $7,797$ tokens after filtering. Since irrelevant tokens hurt LLM reasoning, this reduction should help performance. Second, in an ablation on 20 randomly sampled crashes (10 resolved, 10 unresolved), disabling filtering reduced resolved crashes from 10 to 8, average recall from $0.41$ to $0.35$, and All/Any/None from $34.0/16.0/50.0$ to $29.0/15.0/56.0$. This shows the importance of filtering in *Code Researcher*'s performance.

### 5.5 RQ5: HOW ROBUST ARE *Code Researcher*'S PATCHES?

**Kernel unit tests** In addition to extensive testing (for 10 minutes on 4 machines with 8 parallel processes) for crash-resolution per the kBenchSyz setup, we run kernel unit tests to check if the proposed patches break existing functionality. For each of the 116 crashes resolved by *Code Researcher* (GPT-4o+o1, P@5, 15 max calls), we selected one crash-resolving patch and ran KUnit (Linux, 2025) tests on it. For a total of 28 crashes, either KUnit was not present in the kernel source code version on which the crash was reported or it did not support the required setup. For the remaining 88 crashes, all the KUnit tests had a status of either PASS or SKIP (some hardware-specific tests are skipped depending on the machine requirements). On average, only $\sim 13$ tests were skipped for a patch while $\sim 210$ tests were passed with *no unit test failures reported*.

**Qualitative analysis** While perusing the crash-resolving patches, we came across the following types of patches. Examples for each category (with explanations) are in Listings 1-4, Appendix J. **(1) Accurate** patches correctly identify and fix the root cause of the crash, closely resembling the developer solution. **(2) Overspecialized** patches successfully prevent the crash but may be overspecialized. **(3) Incomplete** patches correctly identify the problem area and approach, but may not be complete. They provide debugging insights and could accelerate the path to a proper fix. **(4) Inaccurate** patches offer a plausible way to resolve the crash, but differ from the developer fix.

### 5.6 RQ6: DOES *Code Researcher* GENERALIZE TO OTHER SYSTEMS CODEBASES?

To demonstrate that *Code Researcher* generalizes with a little effort to other codebases, we experiment with crash resolution in the FFmpeg (FFmpeg, 2025) codebase, a leading open-source multimedia framework. We build a small dataset of 10 recent security-related crash vulnerabilities (which are assigned the top priority) reported by OSS-Fuzz (Google), an automated fuzzing service for open-source projects. Full details of the codebase, dataset construction, and reproduction steps are given in Appendix D. We run *Code Researcher* with the same core prompts as for the Linux kernel. In the unassisted setting (ANALYSIS with GPT-4o, SYNTHESIS with o1, max calls = 15), *Code Researcher* **resolves** 7 **of** 10 **crashes** at Pass@1. It achieves an **average recall of** $0.78$**, editing all the ground-truth files in** 7 **crashes and none in** 2 (excluding one case without a known fix). While FFmpeg crashes are typically not as complex as Linux kernel crashes, our results show that *Code Researcher*'s techniques generalize easily and effectively to other systems codebases.

## 6 CONCLUSIONS, LIMITATIONS, AND FUTURE WORK

In this work, we extend coding agents to deep research scenarios arising in resolving complex issues in large systems codebases. We (a) leverage the rich development history in the codebases (commits), and (b) design effective deep exploration strategies for gathering the rich context often needed to root-cause and patch code crashes. We establish state-of-the-art results on the latest and challenging benchmark of Linux kernel crashes, thoroughly validate our results, perform ablations, and show the generalizability of our approach. Our work currently targets the crash resolution problem, but there are other equally important problems faced by systems software such as slow response times, excessive resource usage and flakiness. It remains to be seen if our deep research strategy could be applied to these scenarios. Deep research for code is a new subfield of agentic AI and we intend to explore novel usecases and strategies beyond the ones presented in the paper.

## REPRODUCIBILITY STATEMENT

We provide all the necessary information to reproduce our results. In Section 4, we include details about our benchmarks and the implementation hyperparameters for *Code Researcher* and all the other baselines. In Appendix G, we give details about our subset of the kBenchSyz benchmark, and provide information about the crash reproduction setup, the unit test setup, compute resources, prompts, and the implementation of the SWE-agent and Agentless baselines. In Appendix D, we give the complete steps to recreate our dataset for FFmpeg and reproduce our results. We also present implementation details for *Code Researcher* in Appendix B. Finally, we submit a supplementary zip file containing (a) the full code of our implementation, (b) prompts and config files used for *Code Researcher* and the SWE-agent and Agentless baselines, (c) both the datasets we used, (d) the patches produced by *Code Researcher* and each baseline, and (e) a README that explains how to reproduce our results.

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

# A ADDITIONAL EXPERIMENTAL RESULTS

**Table of results for Section 5.2** Table 2 shows results that examine whether the tools edit the same files as developers.

Table 2: Average recall and All/Any/None percentages (metrics defined in Section 4) for different tools. LLMs used by the tools are in parentheses.

| Setting | Tool | Max calls | P@k | Avg. Recall | All/Any/None (%) |
|---|---|---|---|---|---|
| Stack context | GPT-4o | 1 | P@5 | **0.46** | **42.5**/7.8/49.7 |
| | o1 | 1 | P@5 | 0.43 | 39.7/7.7/52.6 |
| Unassisted | Agentless (GPT-4o) | 4 | P@5 | 0.49 | 46.4/7.7/45.9 |
| | SWE-agent (GPT-4o) | 15 | P@5 | 0.37 | 35.1/5.6/59.3 |
| | *Code Researcher* **(GPT-4o)** | 15 | P@5 | 0.51 | 48.2/7.8/44.0 |
| | *Code Researcher* **(GPT-4o + o1)** | 15 | P@5 | 0.53 | 49.9/7.6/42.4 |
| | *Code Researcher* **(Gemini 2.5-Flash)** | 15 | P@5 | **0.56** | **52.1**/9.7/38.2 |
| Unassisted + Scaled | SWE-agent (GPT-4o) | 30 | P@5 | 0.40 | 37.9/6.4/55.7 |
| | *Code Researcher* **(GPT-4o)** | 30 | P@5 | **0.53** | **49.5**/8.0/42.5 |
| | SWE-agent (GPT-4o) | 15 | P@10 | 0.36 | 34.3/5.5/60.2 |
| | *Code Researcher* **(GPT-4o)** | 15 | P@10 | 0.51 | 47.8/7.5/44.7 |

**Table of results for Section 5.4** Table 3 shows the results for the `search_commits` ablation on the set of 96 bugs that were successfully resolved by *Code Researcher* (GPT-4o, Pass@5, 15 max calls).

Table 3: Importance of causal analysis of past commits on 96 bugs resolved by *Code Researcher*.

| Tool | Max Calls | P@k | CRR(%) | Avg. Recall | All/Any/None(%) |
|---|---|---|---|---|---|
| *Code Researcher* **(GPT-4o)** | 15 | P@5 | **48.00** | **0.51** | **48.2**/7.8/44.0 |
| W/O `search_commits` | 15 | P@5 | 38.00 | 0.33 | 32.6/2.4/65.0 |

[1] We do this ablation only on the 96 bugs resolved by *Code Researcher* (GPT-4o, Pass@5, 15 max calls).

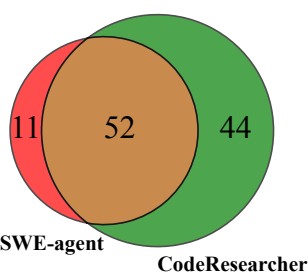

Figure 3: Overlap of crashes resolved by SWE-agent and *Code Researcher* (both at GPT-4o, P@5, 15 max calls)

**Subset analysis of the crashes resolved by *Code Researcher* and SWE-agent** Figure 3 shows a venn diagram of the crashes resolved by *Code Researcher* and SWE-agent in the same configuration (GPT-4o,P@5,15 max calls). It shows that *Code Researcher* is able to solve most of the crashes that SWE-agent resolves, while also resolving a significant number of additional crashes.

## B  *Code Researcher* IMPLEMENTATION DETAILS

The full implementation can be found in the `cresearcher` folder in the supplementary material. We explain a few key points here.

**Context memory**  *Code Researcher* stores all the context collected so far in a context memory. The memory contains a mapping of file path to a list of symbol definitions (further containing fields like symbol name, start position, end position, body, etc.). It also has search queries and results, storing a list of (query, results) pairs where results is itself a list of results. Please see the definition of the `class GlobalCtxAgentState` in the file `cresearcher/utils/types.py` in our supplementary material. In Figure 6 (Appendix C), the context shown provides an example of the memory. We use the memory to form the prompt for each step.

**Prompts**  The system and user prompts for the ANALYSIS and SYNTHESIS phases (for both filtering and patch generation) are provided in the `prompts` folder of our supplementary material. They use a `prompt_preamble` and `prompt_analysis_examples` that vary based on the codebase (e.g., Linux kernel or FFmpeg). Those can be found in the files `config/kBenchSyz.yaml` and `config/ffmpeg.yaml` respectively in our supplementary material. We use the context memory to form the prompt for each step. We show all the open symbol definitions (results of `search_definition` actions), however we only show the results of queries (`search_code` and `search_commits`) from the previous step instead of showing them for all past steps. This is because the results of these queries typically have lower signal to noise ratio and commits can be quite large so they threaten to overwhelm the context. The conversation trajectory is also passed to the LLM, but in case it exceeds our limit of $50K$, we remove intermediate messages as necessary (i.e., the first message containing the crash report and the last message are always kept, and intermediate messages are included as much as possible). In the filtering step (please see `generateGlobalCtxAgentSelectionPrompt` in `cresearcher/globalContext.py` in the supplementary), we prioritize fitting symbol definitions first before we try fitting other search queries and results into our context length limit.

**Implementation of the search actions**  We implement the `search_definition(sym)` action using the `ctags` (universal-ctags) tool to generate (and read) an index file of language objects found in source files for programming languages. The index file is constructed once at the start of *Code Researcher*'s run, usually taking a few minutes for the Linux kernel codebase, and is used throughout the ANALYSIS trajectory. Whenever we show a symbol definition in the prompt, for each line of code that is mentioned in the crash report, we additionally add an annotation (as a C-style comment at the end of the line) saying that this line is important. For `search_code(regex)`, we use the `git grep -E` command to search over all the tracked files in the codebase and show 2 lines of context before and after each matching line. Finally, for `search_commits(regex)`, we use the `git log -E -G` and `git log -E --grep` commands to search over historical commits matching in the code changes and commit messages, respectively. The message and patch of each relevant commit are returned as output, truncated to a maximum of 100 lines. Each action can return a maximum of 5 results. To make these searches over an extremely large repository faster, we progressively search over the files of the symbol definitions present in context memory, then those mentioned in the crash report, then those in the kernel subsystems of the bug, and finally all the files in the codebase. This prioritization strategy allows us to use a timeout of 60 seconds for the `git log` commands (which usually take the longest time) while still getting relevant results in a large number of cases.

**Scalability of search tooling**  We now give some experimental evidence to substantiate our point in Section 2 that existing coding agents that construct repository dependency graphs, like LocAgent (Chen et al., 2025), RepoGraph (Ouyang et al., 2025) and Lingma Agent (Ma et al., 2025), scale poorly to large codebases.

We ran the LocAgent and RepoGraph graph constructions for the `sympy` repository from SWE-bench that contains $\sim 433K$ lines of Python code. LocAgent took 764 seconds. RepoGraph errored out after 44.3 seconds and the progress bar showed `20/1584 [00:24<31:42, 1.22s/it]`, indicating that it would have taken another 30 minutes to complete. In contrast, constructing the `ctags` index used by *Code Researcher* doesn't require any dependency analysis and took 0.76 seconds for the same repository.

## C    EXAMPLE OF AGENT TRAJECTORY

In this section, we show an example ANALYSIS trajectory generated by *Code Researcher* while solving a kernel crash from the dataset[1]. We explain the example here and provide the complete trajectory for reference in Figures 4-10 of Appendix E. The truncated crash report that initiates the investigation of the example is shown in Figure 4.

The trajectory begins in Figure 5. *Code Researcher* identifies a warning in `smsusb_term_device` related to `__flush_work` and forms an initial hypothesis about workqueue synchronization issues. The agent then explores the codebase by examining the `__flush_work` implementation and analyzing the `smsusb_stop_streaming` function's synchronous URB cancellation pattern in Figure 6, while also searching for relevant commit history to identify potential race conditions.

In Figures 7–9, *Code Researcher* traces prior fixes for related bugs in `smsusb_term_device`, inspects synchronization behavior in `cancel_work_sync` and `__cancel_work_timer`. It uncovers a critical misstep where work item initialization via `INIT_WORK` in `smsusb_onresponse` can be bypassed, resulting in NULL `work->func` pointers that explain the observed warning.

Finally, in Figure 10, *Code Researcher* confirms the root cause: though `smsusb_onresponse` correctly initializes work structures with `INIT_WORK`, it is not sufficient. If `smsusb_stop_streaming` is called before any URB completion occurs, the system attempts to cancel uninitialized work items, triggering warnings in `__flush_work` when it encounters NULL function pointers.

## D    FFMPEG: EXPERIMENTAL DETAILS

**Background**    FFmpeg is a leading open-source multimedia framework that supports decoding, encoding, transcoding, muxing, demuxing, streaming, filtering, and playback of virtually all existing media formats. Since it needs to handle a wide range of formats, from very old to the cutting edge, low-level data manipulation is common in the codebase. As of May 2025, FFmpeg has $\sim 4.8K$ files and $\sim 1.46M$ lines of code, primarily in C / C++, with some handwritten assembly for performance.

**Dataset**    We use vulnerabilities discovered by the OSS-Fuzz service (Google) that runs fuzzers on various open source projects and creates alerts for the bugs detected. We focus on security issues, which are assigned the top priority by OSS-Fuzz. These include heap-based buffer overflows, stack-based buffer overflows, use-after-frees, etc. We build a small dataset of 10 FFmpeg crashes, taking the 11 most recent crashes (as of May 14, 2025) that have been verified as fixed and skipping 1 that we could not validate. [2] We use the instructions recommended by OSS-Fuzz for building FFmpeg and testing whether a crash reproduces. [3] The dataset contains the commit at which OSS-Fuzz found the crash, a reproducer file that triggered the crash, and the crash report that we generated by reproducing the crash (the crash reports found by OSS-Fuzz are not publicly visible). We make the dataset available in the `data/ffmpeg` folder in our supplementary material.

**Reproduction**    To run *Code Researcher* on these crashes, we keep the same core prompts (which can be found in the `prompts` folder in the supplementary material), adding a one-paragraph preamble about FFmpeg and replacing the few-shot examples for the Linux kernel with corresponding ones for FFmpeg. The preamble and few-shot examples are in the file `config/ffmpeg.yaml` in our

---

[1]Bug        in        syzkaller:                https://syzkaller.appspot.com/bug?id=
35dd38010e383ee37d27b39f6b50eb25b4c92dd2

[2]https://issues.oss-fuzz.com/issues?q=project:ffmpeg%20type:
vulnerability%20status:verified&s=modified_time:desc&p=1

[3]https://google.github.io/oss-fuzz/advanced-topics/reproducing/

supplementary material. The `README` also contains instructions for running *Code Researcher* on an FFmpeg crash.

## E EXAMPLE AGENT TRAJECTORY: FIGURES

```
smsusb:smsusb_probe: board id=7, interface number 0
------------[ cut here ]------------
WARNING:
CPU: 0 PID: 9 at kernel/workqueue.c:3182 __flush_work+0x95c/0xbf0 kernel/workqueue.c:3182
Call Trace:
 <TASK>
  __cancel_work_timer+0x3e6/0x580 kernel/workqueue.c:3276
  smsusb_stop_streaming drivers/media/usb/siano/smsusb.c:182 [inline]
  smsusb_term_device+0xef/0x300 drivers/media/usb/siano/smsusb.c:344
  smsusb_init_device+0xb60/0xd10 drivers/media/usb/siano/smsusb.c:419
  smsusb_probe+0x5b1/0x10c0 drivers/media/usb/siano/smsusb.c:567
  usb_probe_interface+0x307/0x930 drivers/usb/core/driver.c:396
  ...
  worker_thread+0x687/0x1110 kernel/workqueue.c:2552
  kthread+0x33a/0x430 kernel/kthread.c:379
  ret_from_fork+0x1f/0x30 arch/x86/entry/entry_64.S:308
 </TASK>
```

Figure 4: Crash report of the kernel crash example discussed in Appendix C.

You are analyzing a kernel crash report. Your task is to explore the repository using available tools and gather enough context to completely understand all possible root causes of the crash and how to fix it.

STEP 0 of 9

BUG TITLE : WARNING in `smsusb_term_device`

CRASH REPORT:
{crash_report}

AVAILABLE ACTIONS (write one action per line with no formatting inside <actions> tags):
<actions>
close_definition(filePath, symbolName, startLine)
Close a definition that is not relevant to the bug (e.g. debug functions like dump_stack)
search_definition(filePath, symbolName)
Open the definition of a specified symbol, which can be a function name, a struct name, a constant, a union or a macro
search_definition(symbolName)
Open the definition of a symbol when you don't know its file path. Use this to search for the definition of symbols whose file you don't know.
search_code("regex pattern")
Search through all tracked files in the repository using git grep with regex pattern. Shows 2 context lines before and after each match, limited to 5 matches total.
done                              End exploration because you have gathered enough context to understand the bug and propose a fix. You must only take this action if you are satisfied with the justification you provided in the thoughts section. If unsure, continue exploring. You should also write your justification after this action inside <justification></justification> tags. (followed by justification in <justification> tags)
</actions>
You should examine the evidence and write your thoughts about the current state… Focus on gathering all the relevant context and understanding the bug step-by-step, through careful reasoning steps.

The crash report indicates a warning in the `smsusb_term_device` function. The warning is triggered during a call to `__flush_work` in `kernel/workqueue.c`.
…
The issue appears to be related to the handling of work items, possibly involving improper synchronization or misuse of workqueues. To understand the root cause, I need to examine the `smsusb_term_device` function and the `smsusb_stop_streaming` function to see how work items are managed and flushed. Additionally, I should explore how `__flush_work` is used and what could lead to the warning in this context.

```
<actions>
search_definition("drivers/media/usb/siano/smsusb.c", "smsusb_term_device")
search_definition("drivers/media/usb/siano/smsusb.c", "smsusb_stop_streaming")
search_code("__flush_work")
</actions>
```

The results of actions are added to memory

Figure 5: *Code Researcher* trajectory for the crash described in Figure 4 and discussed in Appendix C: Step 0 of 9

The contents from the memory are added to prompt

STEP 1 of 9

PREVIOUS SEARCHES AND RESULTS:
Query: search_code("__flush_work")
Results:
include/linux/workqueue.h=447=extern bool queue_rcu_work(struct workqueue_struct *wq, struct rcu_work *rwork);
include/linux/workqueue.h-448-
include/linux/workqueue.h:449:extern void __flush_workqueue(struct workqueue_struct *wq);
include/linux/workqueue.h-450-extern void drain_workqueue(struct workqueue_struct *wq);
include/linux/workqueue.h-451-

===

include/linux/workqueue.h=575=extern void __warn_flushing_systemwide_wq(void)
--
include/linux/workqueue.h-615-  if (0)
include/linux/workqueue.h-616-             __warn_flushing_systemwide_wq();                              \
include/linux/workqueue.h:617:  __flush_workqueue(system_wq);                                        \
include/linux/workqueue.h-618-})
include/linux/workqueue.h-619-

( ... more results of this query are not shown in this figure due to space constraints )

CURRENTLY OPEN DEFINITIONS:
File: drivers/media/usb/siano/smsusb.c
Symbol: smsusb_term_device Lines 337 to 355
```c
337| static void smsusb_term_device(...)
338|{
... ( we show the complete definition in the actual prompt )
355| }
```

File: drivers/media/usb/siano/smsusb.c
Symbol: smsusb_stop_streaming
Lines 176 to 189
```c
176| static void smsusb_stop_streaming(...)
177|{
... ( we show the complete definition in the actual prompt )
189| }
```

From the definitions, `smsusb_stop_streaming` iterates over an array of URBs, cancelling each work item synchronously. This synchronous cancellation could be problematic if there's a race condition or if the workqueue is not in a proper state when the function is called.

To understand the root cause, I need to investigate how `cancel_work_sync` is used in conjunction with work items and what specific conditions in `__flush_work` might lead to the warning…. Additionally, I should search for past commits related to `smsusb_term_device`, `smsusb_stop_streaming`, and workqueue handling to see if there have been any recent changes or known issues. …

```
<actions>
search_definition("cancel_work_sync")
search_commits("smsusb_term_device")
search_commits("smsusb_stop_streaming")
search_commits("cancel_work_sync")
</actions>
```

The results of actions are added to memory

Figure 6: *Code Researcher* trajectory for the crash described in Figure 4 and discussed in Appendix C: Step 1 of 9

The contents from the memory are added to prompt

STEP 2 of 9

PREVIOUS SEARCHES AND RESULTS:
Query: `search_commits("smsusb_term_device")`
Results:
Commit message:
commit 31e0456de5be379b10fea0fa94a681057114a96e
Author: Alan Stern <stern@rowland.harvard.edu>
Date: Tue May 7 12:39:47 2019 -0400

media: usb: siano: Fix general protection fault in smsusb…

Patch:
`diff --git a/drivers/media/usb/siano/smsusb.c b/drivers/media/usb/siano/smsusb.c`
`@@ -400,6 +400,7 @@ static int smsusb_init_device(struct usb_interface *intf, int board_id)`
(… we provide the complete commit in the prompt truncated at 100 Lines )

===

Commit message:
786baecfe78f8e25547c628b48a60fc8e5636056
Author: Mauro Carvalho Chehab <mchehab@kernel.org>
Date:   Thu Jun 14 16:35:56 2012 -0300

[media] dvb-usb: move it to drivers/media/usb/dvb-usb…

Patch:
`diff --git a/drivers/media/usb/siano/smsusb.c b/drivers/media/usb/siano/smsusb.c`
(… we provide the complete commit in the prompt truncated to 100 Lines )

( … more results of this and other search_commit query are not shown in this figure due to space constraints )

- - - - - - - - - - - - - - - - - - - - - - - - - - - - - - - - - - - - - - - - - - - - - -

CURRENTLY OPEN DEFINITIONS:
( … the previously opened definitions of `smsusb_term_device` and `smsusb_stop_streaming` are still shown
here in the prompt, but not shown in this figure to save space )

File: kernel/workqueue.c                    Symbol: cancel_work_sync                    Lines 3310 to 3313
```c
3310| bool cancel_work_sync(struct work_struct *work)
3311| {
3312|     return __cancel_work_timer(work, false);
3313| }
```

The commit history reveals some interesting points:
1. There was a fix for a general protection fault in the `smsusb` driver related to endpoint initialization, which
suggests that endpoint management might have been problematic in the past.
2. A past commit addresses use-after-free bugs caused by `do_submit_urb()`, which is relevant since improper
URB management could lead to synchronization issues.
3. Changes to `cancel_work_sync` function suggest potential race conditions or synchronization problems
involving work items.

The definition of `cancel_work_sync` is straightforward, but I need to examine `__cancel_work_timer`

```
<actions>
search_definition("__cancel_work_timer")
search_definition("kernel/workqueue.c", "__flush_work")
</actions>
```

The results of actions are added to memory

Figure 7: *Code Researcher* trajectory for the crash described in Figure 4 and discussed in Appendix C:
Step 2 of 9

The contents from the memory are added to prompt

STEP 3 of 9

CURRENTLY OPEN DEFINITIONS:
( ... The previously opened definitions of `smsusb_term_device`, `smsusb_stop_streaming` and `cancel_work_sync` are still shown here in the prompt )

File: kernel/workqueue.c                    Symbol: __cancel_work_timer                    Lines 3228 to 3290

```c
3228| static bool __cancel_work_timer(struct work_struct *work, bool is_dwork)
3229| {
... ( we show the complete definition in the actual prompt )
3290| }
```

File: kernel/workqueue.c                    Symbol: __flush_work                    Lines 3175 to 3195
```c
3175| static bool __flush_work(struct work_struct *work, bool from_cancel)
3176| {
... ( we show the complete definition in the actual prompt )
3195| }
```

The fact that `__flush_work` is called within `__cancel_work_timer` when the workqueue is online indicates that the warning could be triggered if `work->func` is NULL at this point.

The root cause of the warning likely involves improper initialization or handling of work items, specifically ensuring that the function pointer is set before the work item is queued or flushed. This could be due to a bug in how work items are initialized or how they are managed in the `smsusb_stop_streaming` function.

To confirm this hypothesis, I need to verify how the work items (`dev->surbs[i].wq`) are initialized and assigned their function pointers...

```
<actions>
  search_code("INIT_WORK")
  search_commits("work->func")
  search_commits("smsusb_device_t")
</actions>
```

The results of actions are added to memory

Figure 8: *Code Researcher* trajectory for the crash described in Figure 4 and discussed in Appendix C: Step 3 of 9

The contents from the memory are added to prompt

STEP 4 of 9

PREVIOUS SEARCHES AND RESULTS:
Query: search_code("INIT_WORK")
Results:

drivers/media/usb/siano/smsusb.c=86=static void smsusb_onresponse(struct urb *urb)
--
drivers/media/usb/siano/smsusb.c-144-
drivers/media/usb/siano/smsusb.c-145-exit_and_resubmit:
drivers/media/usb/siano/smsusb.c:146:     INIT_WORK(&surb->wq, do_submit_urb);
drivers/media/usb/siano/smsusb.c:147-     schedule_work(&surb->wq);
drivers/media/usb/siano/smsusb.c-148-}
===
kernel/workqueue.c=520=static inline void debug_work_deactivate(struct work_struct *work)
--
kernel/workqueue.c-523-}
kernel/workqueue.c-524-
kernel/workqueue.c:525:void __init_work(struct work_struct *work, int onstack)
kernel/workqueue.c-526-{
kernel/workqueue.c-527-          if (onstack)

( … more results of this query are not shown in this figure due to space constraints )

| Query: search_commits("work->func") | Query: search_commits("smsusb_device_t") |
|---|---|
| Results: | Results: |
| Commit message: | Commit message: |
| commit c76feb0d5dfdb90b70fa820bb3181142bb01e980 | commit dd47fbd40e6ea6884e295e13a2e50b0894258fdf |
| Author: Paul E. McKenney paulmck@kernel.org | Author: Mauro Carvalho Chehab mchehab@kernel.org |
| workqueue: Make show_pwq() use run-length encoding … | [media] smsusb: don't sleep while atomic… |
| (… we provide the complete commit in the prompt truncated to 100 Lines ) | (… we provide the complete commit in the prompt truncated to 100 Lines ) |
| (… we show the top 5 commits matching the query, not shown here due to space constraints) | (… we show the top 5 commits matching the query, not shown here due to space constraints) |

(…the previously opened definitions are still shown here in the prompt, not included in this figure to save space)

The search results show that the work items (`dev->surbs[i].wq`) are initialized using `INIT_WORK(&surb->wq, do_submit_urb)` in the `smsusb_onresponse` function. The function pointer `do_submit_urb` is assigned during initialization. This means that if the work items are not properly initialized or if the initialization logic is bypassed, the function pointer `work->func` could be NULL, leading to the warning observed in `__flush_work`.…

I should verify the sequence of operations leading to the initialization of work items and their cancellation. …

```
<actions>
search_code("smsusb_onresponse")
search_commits("smsusb_onresponse")
search_commits("smsusb_stop_streaming")
search_commits("race condition")
</actions>
```

The results of actions are added to memory

Figure 9: *Code Researcher* trajectory for the crash described in Figure 4 and discussed in Appendix C: Step 4 of 9

The contents from the memory are added to prompt

STEP 5 of 9

PREVIOUS SEARCHES AND RESULTS:
Query: search_code("smsusb_onresponse")
Results:

drivers/media/usb/siano/smsusb.c=63=static int smsusb_submit_urb(struct smsusb_device_t *dev,
--
drivers/media/usb/siano/smsusb.c-66-/*
drivers/media/usb/siano/smsusb.c-67- * Completing URB's callback handler - bottom half (process context)
drivers/media/usb/siano/smsusb.c:68: * submits the URB prepared on smsusb_onresponse()
drivers/media/usb/siano/smsusb.c-69- */
drivers/media/usb/siano/smsusb.c-70-static void do_submit_urb(struct work_struct *work)
===
drivers/media/usb/siano/smsusb.c-84- * @param urb pointer to a completing urb object
drivers/media/usb/siano/smsusb.c-85- */
drivers/media/usb/siano/smsusb.c:86:static void smsusb_onresponse(struct urb *urb)
drivers/media/usb/siano/smsusb.c-87-{
drivers/media/usb/siano/smsusb.c-88-      struct smsusb_urb_t *surb = (struct smsusb_urb_t *) urb->context;

( ... more results of this query are not shown in this figure due to space constraints )

Query: search_commits("smsusb_onresponse")
Results:
Commit message:
dd47fbd40e6ea6884e295e13a2e50b0894258fdf
Author: Mauro Carvalho Chehab <mchehab@kernel.org>

[media] smsusb: don't sleep while atomic …
(… we provide the complete commit in the prompt truncated at 100 Lines )

(… we show the top 5 commits matching the query, not shown here due to space constraints)

(…The previously opened definitions are still shown here in the prompt, not included in this figure to save space)

Query: search_commits("smsusb_stop_streaming")
Results:
No matches found

Query: search_commits("race condition")
Results:
Commit message:
1d40329736907587ac5995d2d4b3263759ff84e3
Author: Qu Wenruo wqu@suse.com

btrfs: scrub: remove unused path inside scrub_stripe()…
(… we provide the complete commit in the prompt truncated at 100 Lines )

(… we show the top 5 commits matching the query, not shown here due to space constraints)

The `smsusb_onresponse` function is called as a callback for URB completion, and it initializes the work structure with `INIT_WORK(&surb->wq, do_submit_urb)`. … However, if `smsusb_stop_streaming` is called before any URB completes, the work items might be canceled without being initialized, leading to the warning observed in `__flush_work`.
To fix the bug, I need to ensure that work items are only canceled after they have been initialized. This might involve adding checks to ensure URB completion has occurred or modifying the logic to handle uninitialized work items gracefully during cancellation.

I believe I have gathered enough context to understand the bug and propose a fix. The sequence of initialization and cancellation of work items is clear, and the potential for uninitialized work items leading to warnings is evident.

```
<actions>
done
</actions>
```

The results of actions are added to memory

Figure 10: *Code Researcher* trajectory for the crash described in Figure 4 and discussed in Appendix C: Step 5 of 9 ANALYSIS phase

## F    ADDITIONAL RELATED WORK: LONG CONTEXT REASONING

Support for increasing context length sizes in LLMs has been an active area of research (Team et al., 2024; Guo et al., 2023), opening up the possibility of feeding the entire repository into an LLM's context and generating a patch. But there are a few complications. First, note that the Linux kernel has over 75K files and 28 Million lines of code. In contrast, state of the art models today

(e.g., Gemini 2.5 Pro) support at most 2M tokens in the context window (Google, 2025b;c), roughly corresponding to around 100K lines of code (Google, 2025b). Second, long-context models do not robustly make use of the information in context. They often get "lost in the middle" (Liu et al., 2024), performing highest when relevant information occurs at the beginning or end of the input context, and significantly worse when they must access relevant information in the middle of long contexts. Li et al. (2024) found that long-context LLMs struggle with processing long, context-rich sequences and reasoning over multiple pieces of information (which is important for any automated software development task).

## G   EXPERIMENTAL SETUP: ADDITIONAL DETAILS

**Dataset details**   We use the kBenchSyz dataset containing 279 instances from Mathai et al. (2024). The dataset is publicly available at `https://github.com/Alex-Mathai-98/kGym-Kernel-Playground` and is under an MIT License. We validated the 279 instances (i.e., the reproducers and the ground-truth fixes), and ruled out 9 instances for which we could not run the kernel at the parent commit, 27 for which the kernel at the parent commit did not crash, and 43 where the kernel still crashed after applying the fix. So, for our experiments, we use the remaining **200 instances** that we successfully validated. For reproducibility, we use the crash reports generated during our validation instead of the crash reports originally present in kBenchSyz. The subset of 200 instances that we were able to reproduce (containing the bug ids and the crash reports from our reproduction run) is available in the file `data/kBenchSyz/200_subset.json` in the supplementary material.

**Sampling details**   In the SYNTHESIS phase, we ask the agent to generate a hypothesis and patch in the following format. It has to write the hypothesis inside `<hypothesis>` tags and the patch inside `<patch>` tags. The content inside the `<patch>` tags is a list of `<symbol>` tags covering all the symbols whose definitions the agent wants to change in its patch. With each tag, the agent has to provide `file, name and start line` attributes and inside each tag, it has to rewrite the complete definition of the symbol (after making the desired changes). We use successively higher temperatures (0, 0.3, 0.6) until the agent gives a correctly formatted patch. For o1, since its API does not support a temperature parameter, we sample the desired number of patches by setting the `n` parameter (number of completions) in the OpenAI Chat Completions API.

**FFmpeg details**   Please refer to Appendix D for complete details about our experiments on the FFmpeg dataset.

**Crash reproduction setup**   Our setup for building the Linux kernel and running it on reproducer files is built on top of the *kGym* platform (MIT Licensed, publicly available at `https://github.com/Alex-Mathai-98/kGym-Kernel-Gym`) (Mathai et al., 2024) and has a couple of major modifications. First, while *kGym* runs only on the Google Cloud platform, our setup can run locally on any machine and uses cloud storage for preserving compiled kernels, crash reports, etc. Second, we use ccache (ccache) for caching build files generated during kernel compilation and our own logic for caching `git checkouts`.

*kGym* has a distributed setup featuring five workers - *kBuilder*, *kReproducer*, *kScheduler*, *kDashboard* and *kmq*. (1) *kBuilder* takes as input a source commit, a kernel config, and (optionally) a patch. It checks out the kernel at the source commit, applies the patch, compiles the kernel and uploads the build artifacts (kernel image, vmlinux binary, etc.) to cloud storage. (2) *kReproducer* takes as input the build artifacts and a reproducer file and runs the kernel on the reproducer while monitoring for crashes. To handle non-deterministic bugs, we launch 4 VMs in parallel, each of which runs the reproducer. Each VM further runs multiple processes where system calls can execute in parallel so concurrency bugs can also be reproduced. If any of these VMs crash within 10 minutes or if *kReproducer* loses connection to the VMs, we say that the kernel crashes on the reproducer. It then uploads the crash reports to cloud storage. (3) *kScheduler* serves an API where we can send reproduction jobs with the source commit, config, reproducer and (optionally) patch. It communicates with *kBuilder* and *kReproducer* through the message queue *kmq* and orchestrates the overall flow of build with *kBuilder* followed by reproduction with *kReproducer*. (4) Finally, *kDashboard* displays each job's logs and results in a web UI.

**KUnit testing details** For each of the 116 crashes resolved by *Code Researcher* (GPT-4o+o1, P@5, 15 max calls), we selected one crash-resolving patch and ran KUnit (Linux, 2025) tests on it. For 13 crashes, KUnit was not present in the kernel source code version on which the crash was reported. For another 15 crashes, KUnit was present but did not support the CLI arguments for running tests in QEMU (required as our host kernel is different from the test kernel). From the remaining 88 crashes, all the KUnit tests had a status of either PASS or SKIP (some hardware-specific tests are skipped depending on the machine requirements). On average, only $\sim 13$ tests were skipped for a patch while $\sim 210$ tests were passed. Due to a bug in the KUnit test-suite at the parent of the fix commit for 4 crashes, the example_skip_test and example_mark_skipped_test tests, which should have been skipped, were run. Similarly, for 2 crashes, due to a bug in KUnit[4], hw_breakpoint tests, that should have been skipped, were run. We ignore the results of these tests as they are not relevant to the correctness of the patches being tested.

**Compute resources** We setup 10 replicas of the distributed setup (containing 5 workers) described above. Each machine was equipped with an AMD EPYC 7V13 Processor running at 2.50 GHz, had 24 cores and 220 GB RAM. For one evaluation run on our dataset of 200 instances for any tool in the P@5 setting (i.e., for evaluating 1000 patches on whether they prevent a crash or not), we divided the instances among the 10 replicas, and the overall time ranged from 10 to 15 hours.

*Code Researcher* **Prompts** The system and user prompts for the ANALYSIS and SYNTHESIS phases (for both filtering and patch generation) are provided in the prompts folder of our supplementary material. They use a prompt_preamble and prompt_analysis_examples that vary based on the codebase (e.g., Linux kernel or FFmpeg). Those can be found in the files config/kBenchSyz.yaml and config/ffmpeg.yaml respectively in our supplementary material.

**SWE-agent details** We use SWE-agent (Yang et al., 2024) as one of our baselines. The codebase is publicly available at https://github.com/SWE-agent/SWE-agent/tree/main and is under the MIT License. We use version 1.0.1 of SWE-agent, and add a Linux kernel-specific example trajectory and background about the Linux kernel to its prompts. We provide the complete configuration file (including all prompts and the example trajectory) in the SWE-agent folder of our supplementary material.

**Agentless details** We adopt Agentless (Xia et al., 2024) as one of our baselines. The codebase is publicly available at https://github.com/OpenAutoCoder/Agentless and is under MIT License. The default Agentless pipeline consists of the following steps: (i) retrieval of two sets of relevant files (via an LLM and via embeddings), (ii) identification of candidate edit locations, (iii) generation of multiple patch candidates, and (iv) ranking of patches using test executions. In our implementation, we fix the temperature at 0.1 for all stages of the pipeline and sample 5 candidate patches in the third stage of patch generation. For scalability reasons, we omit the embedding-based retrieval step and ranking is immaterial in our setting since we use Pass@5. Although the original implementation could not be directly reused, since it is designed specifically for Python codebases, we reimplemented the stages of the pipeline for our usecase, incorporating kernel-specific prompts at each step.

**Rationale for omitting embedding-based retrieval** To estimate the computational overhead, we randomly sampled $1,000$ .c and .h files from the Linux kernel and computed the average number of chunks per file. Extrapolating to the entire codebase, we estimate approximately $781,889$ chunks. With the text-embedding-3-small model (used by Agentless), each embedding call requires on average $\sim 0.5$ seconds. This implies a total embedding time of roughly 108 hours for a single kernel snapshot, making this step infeasible in our setting.

All prompts used for the Agentless stages in our experiments are included in the Agentless folder of the supplementary material.

---

[4]See https://groups.google.com/g/kunit-dev/c/ahWFBJsIA2U and https://www.spinics.net/lists/kernel/msg5128827.html.

## H  EXAMPLE SHOWING THE IMPORTANCE OF CAUSAL ANALYSIS OVER HISTORICAL COMMITS

**Figures 11–13:** Illustration of *Code Researcher* analyzing and repairing a real-world memory leak bug[5] from the kBenchSyz dataset (the complete trajectory of *Code Researcher* is truncated and only the relevant parts are shown due to space constraints). Figure 11 shows the developer's original commit, including the fix and a "Fixes:" tag that references the buggy commit where the issue originated: commit `6679f4c5e5a6`—highlighted in yellow. This section is shown in the orange box . The developer's fix is available at the following link[6]. Figure 12 displays a subset of actions taken by *Code Researcher* in the ANALYSIS phase, specifically several `search_commits` steps ( green box ), which reveal how the agent retrieves both the buggy commit and other related commits that involve memory management. These retrieved historical commits provide guidance in the SYNTHESIS phase. Figure 13 presents the patch and analysis synthesized by *Code Researcher* ( blue box ) during the SYNTHESIS phase, which correctly identifies the missing deallocation and inserts the appropriate `kfree(bt_const_extended)` call. Notably, *Code Researcher* successfully navigates to the same buggy commit identified by the developer's "Fixes:" tag, demonstrating its ability to infer causality and leverage prior commits to localize and fix bugs.

```
commit   50d34a0
author   Marc Kleine-Budde
Fixes    6679f4c5e5a6 ("can: gs_usb: add extended bt_const feature")
```

…The information must be read with an USB control message. The memory for this message is allocated but ***not free()ed*** (in the non error case). ***This patch adds the missing free.***

```
<patch>
static struct gs_can *gs_make_candev(unsigned int channel,
   dev->data_bt_const.brp_inc = le32_to_cpu(bt_const_extended->dbrp_inc);
   dev->can.data_bittiming_const = &dev->data_bt_const;
+
+    kfree(bt_const_extended);
  }

   SET_NETDEV_DEV(netdev, &intf->dev);
</patch>
```

**Developer commit message and patch**

Figure 11: Developer commit message and patch.

---

[5]Bug in Syzbot dashboard: https://syzkaller.appspot.com/bug?id=92a742e993c8b9e769f8502a0497c88c0afa78af.

[6]Developer's fix commit: https://git.kernel.org/pub/scm/linux/kernel/git/torvalds/linux.git/commit/?id=50d34a0d151dc7abbdbec781bd7f09f2b3cbf01a.

**Actions taken by Code Researcher**

🔍 **search_commits("bt_const_extended|memory leak")**
commit 6679f4c5e5a6
Author: Peter Fink

can: gs_usb: add extended bt_const feature

… phase to reach maximum    performance at higher speeds. …
```
<patch>
static struct gs_can *gs_make_candev(unsigned int channel,
    struct net_device *netdev;
    int rc;
    struct gs_device_bt_const *bt_const;
+   struct gs_device_bt_const_extended *bt_const_extended; u32 feature;
    bt_const = kmalloc(sizeof(*bt_const), GFP_KERNEL);
</patch>
```

🔍 **search_commits("kfree|memory management")**
commit 5cda3ee5
Author: Marc Kleine-Budde
Date:   Sun Jun 4 14:03:42 2017 +0200

can: gs_usb: fix memory leak in gs_cmd_reset()

***This patch adds the missing kfree()*** in gs_cmd_reset() to free the    memory that is not used anymore after usb_control_msg().

```
<patch>
static int gs_cmd_reset(struct gs_usb *gsusb, struct gs_can *gsdev)
    sizeof(*dm),
    1000);
+   kfree(dm);
+
    return rc;
}
</patch>
```

Figure 12: *Code Researcher* actions (`search_commits` in green box).

The crash report indicates a memory leak in the `gs_usb_probe` function… ***The search results and past commits provide examples of similar memory management issues*** …The patch ***should add a kfree*** call for `bt_const_extended` after its data has been used to populate the `data_bt_const` fields in the `gs_make_candev` function.

```
<patch>
static struct gs_can *gs_make_candev(unsigned int channel,
      dev->data_bt_const.brp_inc = le32_to_cpu(bt_const_extended->dbrp_inc);

      dev->can.data_bittiming_const = &dev->data_bt_const;
+
+      kfree(bt_const_extended); // Free the allocated memory for bt_const_extended
   }

   SET_NETDEV_DEV(netdev, &intf->dev);
</patch>
```

**Analysis and patch by Code Researcher**

Figure 13: *Code Researcher* patch and analysis.

# I LLM-AS-JUDGE EVALUATION OF OVERLAP BETWEEN DEVELOPER COMMIT AND TOOL-GATHERED CONTEXT

We use LLM-as-judge to analyze the context gathered by *Code Researcher* and SWE-agent to determine the overlap of context in their trajectory with the context mentioned by the developer in the ground-truth fix commit message. We first identify code symbols mentioned in the commit message for a given bug $b$, which we denote as $s_b^*$. Then for each candidate patch $i$, we find the overlap of $s_b^*$ with the symbols whose definitions are seen in its trajectory. We denote this overlap by $s_{b,i}$. We define symbol ratio $SR$ for each candidate patch as

$$SR_{b,i} = \frac{|s_{b,i}|}{|s_b^*|}.$$

We consider patch $i$ to have overlapping symbol context with the developer commit if $SR_{b,i} \geq 0.33$. We label all candidate patches with this criterion. As mentioned in Section 5.3 (2), we find that SWE-agent has $54.18\%$ overlapping symbol context patches, while *Code Researcher* has $63.7\%$ overlapping symbol context patches. This indicates that *Code Researcher* is more effective at identifying relevant context.

Additionally, we also measure the impact of finding relevant context on the crash resolution rate (CRR) as:

$$P(\text{patch resolves crash} \mid \text{overlapping symbol context}) = 0.309,$$

$$P(\text{patch resolves crash} \mid \text{non-overlapping symbol context}) = 0.116.$$

This suggests that patches with overlapping symbol context have a significantly higher probability of resolving crashes than patches without.

In addition to symbols, we also identify commit IDs mentioned in the commit message for a given bug $b$ which we denote as $c_b^*$. Then for each candidate patch $i$, we find the overlap of $c_b^*$ with the commits retrieved in its trajectory, denoted as $c_{b,i}$. We note that $c_b^*$ is typically a small number, with a maximum value of 3 in our dataset of 200 bugs. Therefore, instead of a ratio, we label patch $i$ to have overlapping commit context when all the commits in $c_b^*$ are present in $c_{b,i}$ (i.e., $\frac{|c_{b,i}|}{|c_b^*|} = 1$). We find that $30.8\%$ of patches produced by *Code Researcher* have overlapping commit context (recall that SWE-agent does not search over commit IDs). Further, we find that overlapping commit context also has a positive impact on CRR:

$$P(\text{patch resolves crash} \mid \text{overlapping commit context}) = 0.315,$$

$$P(\text{patch resolves crash} \mid \text{non-overlapping commit context}) = 0.205.$$

Overall, these results bring out the utility of effective context retrieval.

## J   QUALITATIVE EVALUATION AND EXAMPLES

**Example A: `jfs_dmap.c` boundary check.**   Listing 1[7] compares the developer's ground-truth patch with the patch generated by *Code Researcher*. Both fixes add a lower-bound check on `bmp->db_agl2size` alongside the existing upper-bound check; the only difference is the ordering of the two disjuncts in the `if` condition, an immaterial variation in this case. This illustrates the class of *Accurate* patches.

Listing 1: Semantically equivalent patch produced by *Code Researcher* for the jfs_dmap.c crash.

```
1  --- a/fs/jfs/jfs_dmap.c                                 /* developer */
2  +++ b/fs/jfs/jfs_dmap.c
3  @@ -193,7 +193,8 @@ int dbMount(struct inode *ipbmap)
4      bmp->db_agwidth = le32_to_cpu(dbmp_le->dn_agwidth);
5      bmp->db_agstart = le32_to_cpu(dbmp_le->dn_agstart);
6      bmp->db_agl2size = le32_to_cpu(dbmp_le->dn_agl2size);
7  -    if (bmp->db_agl2size > L2MAXL2SIZE - L2MAXAG) {
8  +    if (bmp->db_agl2size > L2MAXL2SIZE - L2MAXAG ||
9  +        bmp->db_agl2size < 0) {
10         err = -EINVAL;
11         goto err_release_metapage;
12     }
13
14 --- a/fs/jfs/jfs_dmap.c                                 /* generated */
15 +++ b/fs/jfs/jfs_dmap.c
16 @@ -193,7 +193,7 @@ int dbMount(struct inode *ipbmap)
17     bmp->db_agwidth = le32_to_cpu(dbmp_le->dn_agwidth);
18     bmp->db_agstart = le32_to_cpu(dbmp_le->dn_agstart);
19     bmp->db_agl2size = le32_to_cpu(dbmp_le->dn_agl2size);
20 -    if (bmp->db_agl2size > L2MAXL2SIZE - L2MAXAG) {
21 +    if (bmp->db_agl2size < 0 ||  bmp->db_agl2size > L2MAXL2SIZE
   ↪  - L2MAXAG) {
22         err = -EINVAL;
23         goto err_release_metapage;
24     }
```

**Example B: `hci_h5.c` null-check addition.**   In Listing 2[8], both the developer and *Code Researcher* address an unsafe access to `hu->serdev->dev`, but the generated patch goes beyond the developer's fix. While the developer simply guards the power management calls with a null check, *Code Researcher* adds an `else` branch that logs an error and returns `-ENODEV`. This reflects a conservative design that prevents execution in the event of a null pointer, even though the surrounding kernel code may already guarantee that `hu->serdev` is non-null. Such overspecialization can be seen as a benign deviation: the patch is functionally correct and improves robustness, but at the risk of silently diverging from upstream assumptions. This illustrates the class of *overspecialized* patches.

Listing 2: Overspecialized patch generated for hci_h5.c.

```
1  --- a/drivers/bluetooth/hci_h5.c                         /* developer */
2  +++ b/drivers/bluetooth/hci_h5.c
3  @@ -587,9 +587,11 @@ static int h5_recv(...)
4      count -= processed;
5      }
6
7  -    pm_runtime_get(&hu->serdev->dev);
8  -    pm_runtime_mark_last_busy(&hu->serdev->dev);
9  -    pm_runtime_put_autosuspend(&hu->serdev->dev);
```

---

[7]Example A in Syzkaller: https://syzkaller.appspot.com/bug?id=5eb8a5d29d77f8a364cf3270bf9625eb4d4ffc52

[8]Example B in Syzkaller: https://syzkaller.appspot.com/bug?id=1bc76a5779300075e3466a0e2abcd753df212779

```
10 +    if (hu->serdev) {
11 +        pm_runtime_get(&hu->serdev->dev);
12 +        pm_runtime_mark_last_busy(&hu->serdev->dev);
13 +        pm_runtime_put_autosuspend(&hu->serdev->dev);
14 +    }
15
16      return 0;
17  }
18
19
20 --- a/drivers/bluetooth/hci_h5.c                    /* generated */
21 +++ b/drivers/bluetooth/hci_h5.c
22 @@ -587,9 +587,14 @@ static int h5_recv(...)
23                  count -= processed;
24          }
25
26 -    pm_runtime_get(&hu->serdev->dev);
27 -    pm_runtime_mark_last_busy(&hu->serdev->dev);
28 -    pm_runtime_put_autosuspend(&hu->serdev->dev);
29 +    if (hu->serdev) {
30 +        pm_runtime_get(&hu->serdev->dev);
31 +        pm_runtime_mark_last_busy(&hu->serdev->dev);
32 +        pm_runtime_put_autosuspend(&hu->serdev->dev);
33 +    } else {
34 +        bt_dev_err(hu->hdev, "serdev is not initialized");
35 +        return -ENODEV;
36 +    }
37
38      return 0;
39  }
```

**Example C: `ns.c` RCU read lock insertion.** In Listing 3[9] both the developer and *Code Researcher* address the unsafe traversal of a radix tree without proper RCU synchronization. The developer applies a comprehensive fix, wrapping all relevant `radix_tree_for_each_slot` iterations with `rcu_read_lock()` and `rcu_read_unlock()` across multiple functions. In contrast, *Code Researcher* focuses only on the `ctrl_cmd_new_lookup()` function, inserting the necessary locking primitives in that scope alone. While this partial patch is not directly mergeable due to its incompleteness, it demonstrates an accurate understanding of the underlying concurrency issue and correctly applies the mitigation in the context it modifies. As such, it exemplifies the class of *incomplete* patches, offering concrete insight into the nature and location of the bug, and accelerating the path toward a complete and upstreamable fix.

Listing 3: Developer and plausible patches for ns.c.

```
1 --- a/net/qrtr/ns.c                                /* developer */
2 +++ b/net/qrtr/ns.c
3 @@ -193,12 +193,13 @@ static int announce_servers(struct
  ↪   sockaddr_qrtr *sq)
4          struct qrtr_server *srv;
5          struct qrtr_node *node;
6          void __rcu **slot;
7 -        int ret;
8 +        int ret = 0;
9
10         node = node_get(qrtr_ns.local_node);
11         if (!node)
12                 return 0;
13
```

---

[9]Example C in Syzkaller: https://syzkaller.appspot.com/bug?id=07c9d71dc1a215b19c6a245c68f502bc57dbdb83

```
14  +           rcu_read_lock();
15              /* Announce the list of servers registered in this
                ↪   node */
16              radix_tree_for_each_slot(slot, &node->servers, &iter,
                ↪   0) {
17                      srv = radix_tree_deref_slot(slot);
18  @@ -206,11 +207,14 @@ static int announce_servers(struct
    ↪   sockaddr_qrtr *sq)
19                      ret = service_announce_new(sq, srv);
20                      if (ret < 0) {
21                              pr_err("failed to announce new
                                ↪   service\n");
22  -                           return ret;
23  +                           goto err_out;
24                      }
25              }
26
27  -           return 0;
28  +err_out:
29  +           rcu_read_unlock();
30  +
31  +           return ret;
32   }
33
34   static struct qrtr_server *server_add(unsigned int service,
35  @@ -335,7 +339,7 @@ static int ctrl_cmd_bye(struct
    ↪   sockaddr_qrtr *from)
36              struct qrtr_node *node;
37              void __rcu **slot;
38              struct kvec iv;
39  -           int ret;
40  +           int ret = 0;
41
42              iv.iov_base = &pkt;
43              iv.iov_len = sizeof(pkt);
44  @@ -344,11 +348,13 @@ static int ctrl_cmd_bye(struct
    ↪   sockaddr_qrtr *from)
45              if (!node)
46                      return 0;
47
48  +           rcu_read_lock();
49              /* Advertise removal of this client to all servers of
                ↪   remote node */
50              radix_tree_for_each_slot(slot, &node->servers, &iter,
                ↪   0) {
51                      srv = radix_tree_deref_slot(slot);
52                      server_del(node, srv->port);
53              }
54  +           rcu_read_unlock();
55
56              /* Advertise the removal of this client to all local
                ↪   servers */
57              local_node = node_get(qrtr_ns.local_node);
58  @@ -359,6 +365,7 @@ static int ctrl_cmd_bye(struct
    ↪   sockaddr_qrtr *from)
59              pkt.cmd = cpu_to_le32(QRTR_TYPE_BYE);
60              pkt.client.node = cpu_to_le32(from->sq_node);
61
62  +           rcu_read_lock();
```

```
63              radix_tree_for_each_slot(slot, &local_node->servers,
                ↪  &iter, 0) {
64                      srv = radix_tree_deref_slot(slot);
65
66 @@ -372,11 +379,14 @@ static int ctrl_cmd_bye(struct
   ↪  sockaddr_qrtr *from)
67                      ret = kernel_sendmsg(qrtr_ns.sock, &msg, &iv,
                        ↪  1, sizeof(pkt));
68                      if (ret < 0) {
69                              pr_err("failed to send bye cmd\n");
70 -                            return ret;
71 +                            goto err_out;
72                      }
73              }
74
75 -        return 0;
76 +err_out:
77 +        rcu_read_unlock();
78 +
79 +        return ret;
80  }
81
82  static int ctrl_cmd_del_client(struct sockaddr_qrtr *from,
83 @@ -394,7 +404,7 @@ static int ctrl_cmd_del_client(struct
   ↪  sockaddr_qrtr *from,
84          struct list_head *li;
85          void __rcu **slot;
86          struct kvec iv;
87 -        int ret;
88 +        int ret = 0;
89
90          iv.iov_base = &pkt;
91          iv.iov_len = sizeof(pkt);
92 @@ -434,6 +444,7 @@ static int ctrl_cmd_del_client(struct
   ↪  sockaddr_qrtr *from,
93          pkt.client.node = cpu_to_le32(node_id);
94          pkt.client.port = cpu_to_le32(port);
95
96 +        rcu_read_lock();
97          radix_tree_for_each_slot(slot, &local_node->servers,
                ↪  &iter, 0) {
98                  srv = radix_tree_deref_slot(slot);
99
100 @@ -447,11 +458,14 @@ static int ctrl_cmd_del_client(struct
    ↪  sockaddr_qrtr *from,
101                     ret = kernel_sendmsg(qrtr_ns.sock, &msg, &iv,
                       ↪  1, sizeof(pkt));
102                     if (ret < 0) {
103                             pr_err("failed to send del client
                               ↪  cmd\n");
104 -                           return ret;
105 +                           goto err_out;
106                     }
107             }
108
109 -        return 0;
110 +err_out:
111 +        rcu_read_unlock();
112 +
113 +        return ret;
```

```
114  }
115
116  static int ctrl_cmd_new_server(struct sockaddr_qrtr *from,
117  @@ -554,6 +568,7 @@ static int ctrl_cmd_new_lookup(struct
     ↪  sockaddr_qrtr *from,
118          filter.service = service;
119          filter.instance = instance;
120
121 +        rcu_read_lock();
122          radix_tree_for_each_slot(node_slot, &nodes, &node_iter,
            ↪  0) {
123                  node = radix_tree_deref_slot(node_slot);
124
125  @@ -568,6 +583,7 @@ static int ctrl_cmd_new_lookup(struct
     ↪  sockaddr_qrtr *from,
126                          lookup_notify(from, srv, true);
127                  }
128          }
129 +        rcu_read_unlock();
130
131          /* Empty notification, to indicate end of listing */
132          lookup_notify(from, NULL, true);
133
134
135 --- a/net/qrtr/ns.c                      /* generated */
136 +++ b/net/qrtr/ns.c
137  @@ -554,6 +554,7 @@ static int ctrl_cmd_new_lookup(struct
     ↪  sockaddr_qrtr *from,
138          filter.service = service;
139          filter.instance = instance;
140
141 +        rcu_read_lock(); // Add RCU read lock
142          radix_tree_for_each_slot(node_slot, &nodes, &node_iter,
            ↪  0) {
143                  node = radix_tree_deref_slot(node_slot);
144
145  @@ -568,6 +569,7 @@ static int ctrl_cmd_new_lookup(struct
     ↪  sockaddr_qrtr *from,
146                          lookup_notify(from, srv, true);
147                  }
148          }
149 +        rcu_read_unlock(); // Add RCU read unlock
150
151          /* Empty notification, to indicate end of listing */
152          lookup_notify(from, NULL, true);
```

**Example D: `qrtr.c` port validation.** In Listing 4[10], the developer replaces `idr_alloc()` with `idr_alloc_u32()` to avoid casting the (possibly large) `u32` port number to `int`. By contrast, *Code Researcher* adds defensive checks that reject ports with `port < 0`, both in `qrtr_port_assign` and `__qrtr_bind`. This patch resolves the crash, but rejects certain port numbers rather than handling them, so is not equivalent to the developer patch and is *inaccurate*. But the incoming value originates from `__u32 sq_port`, and special constants like `QRTR_PORT_CTRL` (defined as `0xffffffffu`) are valid and widely used in the subsystem.

Listing 4: Developer and inaccurate patches for `qrtr.c`.

```
1 --- a/net/qrtr/qrtr.c                                /* developer */
2 +++ b/net/qrtr/qrtr.c
```

---

[10]Example D in Syzkaller: https://syzkaller.appspot.com/bug?id=ca2299cf11b3e3d3d0f44ac479410a14eecbd326

```
3  @@ -692,23 +692,25 @@ static void qrtr_port_remove(struct
   ↪  qrtr_sock *ipc)
4   */
5  static int qrtr_port_assign(struct qrtr_sock *ipc, int *port)
6  {
7 +        u32 min_port;
8         int rc;
9
10        mutex_lock(&qrtr_port_lock);
11        if (!*port) {
12 -               rc = idr_alloc(&qrtr_ports, ipc,
13 -                             QRTR_MIN_EPH_SOCKET,
   ↪  QRTR_MAX_EPH_SOCKET + 1,
14 -                             GFP_ATOMIC);
15 -               if (rc >= 0)
16 -                       *port = rc;
17 +               min_port = QRTR_MIN_EPH_SOCKET;
18 +               rc = idr_alloc_u32(&qrtr_ports, ipc, &min_port,
   ↪  QRTR_MAX_EPH_SOCKET, GFP_ATOMIC);
19 +               if (!rc)
20 +                       *port = min_port;
21        } else if (*port < QRTR_MIN_EPH_SOCKET &&
          ↪  !capable(CAP_NET_ADMIN)) {
22                rc = -EACCES;
23        } else if (*port == QRTR_PORT_CTRL) {
24 -               rc = idr_alloc(&qrtr_ports, ipc, 0, 1,
   ↪  GFP_ATOMIC);
25 +               min_port = 0;
26 +               rc = idr_alloc_u32(&qrtr_ports, ipc, &min_port,
   ↪  0, GFP_ATOMIC);
27        } else {
28 -               rc = idr_alloc(&qrtr_ports, ipc, *port, *port
   ↪  + 1, GFP_ATOMIC);
29 -               if (rc >= 0)
30 -                       *port = rc;
31 +               min_port = *port;
32 +               rc = idr_alloc_u32(&qrtr_ports, ipc, &min_port,
   ↪  *port, GFP_ATOMIC);
33 +               if (!rc)
34 +                       *port = min_port;
35        }
36        mutex_unlock(&qrtr_port_lock);
37
38 --- a/net/qrtr/qrtr.c                    /* generated */
39 +++ b/net/qrtr/qrtr.c
40 @@ -695,7 +695,9 @@ static int qrtr_port_assign(struct
   ↪  qrtr_sock *ipc, int *port)
41        int rc;
42
43        mutex_lock(&qrtr_port_lock);
44 -        if (!*port) {
45 +        if (*port < 0) {
46 +                rc = -EINVAL;
47 +        } else if (!*port) {
48                rc = idr_alloc(&qrtr_ports, ipc,
49                              QRTR_MIN_EPH_SOCKET,
                              ↪  QRTR_MAX_EPH_SOCKET + 1,
50                              GFP_ATOMIC);
51 @@ -759,6 +761,9 @@ static int __qrtr_bind(struct socket *sock,
52                        return 0;
```

```
53
54          port = addr->sq_port;
55 +        if (port < 0)
56 +                return -EINVAL;
57 +
58          rc = qrtr_port_assign(ipc, &port);
59          if (rc)
60                  return rc;
```

