# OpenReview forum: "Code Researcher: Deep Research Agent for Large Systems Code and Commit History"
_ICLR.cc/2026/Conference — Submitted to ICLR 2026_

### Official Review · Reviewer_6PPQ · 2025-10-27

**Soundness:** 3
**Presentation:** 3
**Contribution:** 3
**Rating:** 6
**Confidence:** 4

**Summary:**

This paper presents a Deep Research Agent on Code, specifically applying it to a problem set of generating patches to mitigate crashes in systems code. The agent works in three phases of (1) Analysis, (2) Synthesis; and (3) Validation; and the agent is evaluated on a chanllenging benchmark kBenchSyz. Experimental results demonstrate superior advantage of the agent capable of outperforming prior works on gathering relevant context.

**Strengths:**

(1) The work targets real world complex benchmarks, including large system repository which requires advanced reasoning and context search.

(2) The authors leveraged new action tools, especially allowing the Agent to search through commit histories, to gather enough relevant context related to systems crash and leverage advanced reasoning strategies related to the problem such as chasing control and data flow chains.

(3) Results demonstrate superior performance compared with SweAgent, especially on the global context gathering capability of the proposed agent.

**Weaknesses:**

(1) Results in Table 1 could be difficult to comprehend. Specifically why are some settings using P@5, P@10 and P@16? This makes comparing between different settings/tools difficult. Why not just use P@1?

(2) How test time scaling was conducted was not properly explained and the results are not analyzed deeply.

(3) It would be great if results are also presented for SweBench.

**Questions:**

Please refer to weaknesses.

---

> ### Author Response · Authors · 2025-11-21
>
> Thank you for your comments and for acknowledging the complexity of our problem and the importance of commit history. We address each of the weaknesses below.
>
> 1. **Choice of Settings in Table 1**: We would like to clarify that for all the experiments we have done (except for the test-time scaling experiments where we have studied increasing P@5 to P10), we have evaluated both Code Researcher and all baselines in the P@5 setting. For CrashFixer, since the code is not open-sourced, we could not run any experiments and could only present the numbers mentioned in their paper. Please note that this evaluation, in fact, gives an unfair advantage to CrashFixer. This is because (i) CrashFixer is evaluated in the assisted setting where the buggy files are already given while Code Researcher is just given the crash report and (ii) CrashFixer uses a higher sampling budget of Pass@16, as compared to the Pass@5 budget used by Code Researcher.
>
> 2. **Analysis of Test-Time Scaling**: We evaluated test-time scaling along two axes, (i) increasing the number of agent trajectories (and thus patches generated), and (ii) allowing the agent to continue trajectories for more steps. Keeping the total number of LLM calls the same, we increased both these parameters and found that increasing the number of trajectories helps while increasing their length doesn't help much. You are right, this indeed merits some more investigation. We analysed some of Code Researcher's trajectories on crashes it couldn't resolve with more steps but was able to with more trajectories. We found that if it picks a wrong line of investigation (e.g., thinking the fault is inside a function when it is with the arguments being passed to the function), it finds it hard to backtrack. While there is some room for changing its beliefs, it is unlikely that a completely different line of thought is explored. So it is more effective to explore multiple independent lines of thought rather than relying on more steps to correct an existing line of thought. We will try to add this analysis and improve our explanation in the paper, thank you for the suggestion.
>
> 3. **Results on SWE-bench**: Thank you for the suggestion. However, we are unable to understand why an experiment on SWE-bench would be relevant for our approach. SWE-bench consists of application level repositories where coding agents have already achieved a very high success rate despite the fact that none of them make use of the commit history (which, as we have shown, is quite important for large codebases). Coding agents designed for SWE-bench iteratively make edits, test their changes, make more edits, and so on. Our agent is designed for large and complex codebases where (i) building and testing may be infeasible to do in an agent loop and (ii) the complexity requires gathering lots of global context *before* making any changes. Thus, SWE-bench is a fundamentally different problem setting than the one we aim to tackle in our paper. The difference in the settings is further highlighted by the fact that SWE-agent, a SOTA agent on SWE-bench, is only able to resolve $37.5$% of crashes while Code Researcher resolves $48$%, keeping everything else the same.

---

### Official Review · Reviewer_SMcB · 2025-10-30

**Soundness:** 3
**Presentation:** 3
**Contribution:** 2
**Rating:** 4
**Confidence:** 3

**Summary:**

This paper introduces Code Researcher, an LLM-based agent workflow designed to resolve crash issues in complex system-level codebases. Under comparable settings with GPT-4o, Code Researcher achieves a 48% crash-resolution rate, outperforming common baselines such as SWE-Agent and Agentless.

**Strengths:**

- To the best of my knowledge, this appears to be the first work explicitly exploring deep research within software engineering tasks, which makes it novel in scope.

- The paper tackles realistic and challenging problems—specifically, software development and maintenance tasks in complex Linux kernel C/C++ codebases. Compared to prior work focusing on simpler scripting languages such as Python (e.g., SWE-bench), the chosen domain is harder, more practical, and less explored.

**Weaknesses:**

- I find the claim of performing “Deep Research for Code” somewhat confusing.
From my understanding, project-level issue resolution inherently requires agentic search and information summarization. Thus, the conceptual gap between Code Researcher and a standard coding agent seems much smaller than the gap between a general AI chatbot and a deep research agent.
Could the authors clarify what fundamental difference distinguishes a Code Researcher from a regular coding agent? Specifically, what unique design elements justify the term “deep research”? Simply shifting from small Python projects to large-scale kernel-level issues or using commit history do not make an agent a deep researcher; there should be distinctive mechanisms or reasoning processes that set it apart.

- According to the authors’ claim, the main challenges of kernel code issue lie in the large scale and complexity of the codebase. However, the design of Code Researcher does not appear to include any distinctive mechanisms specifically aimed at addressing this challenge. The code search tools are already used in most existing SWE agents, making it unclear what unique architectural or algorithmic features enable Code Researcher to outperform existing agents on kernel-level code. It is also worth noting that a better prompt alone does not constitute a research contribution.

- The memory filtering mechanism in the Synthesis stage is particularly intriguing. This may be one of the few substantial architectural differences from existing coding agents, which typically include retrieval, recall, and patch generation stages.
However, the paper does not include any ablation studies or quantitative analysis demonstrating the contribution of this filtering design. Without such evidence, it’s unclear how much this component actually contributes to the overall performance.

- The dataset used seems potentially affected by the incomplete patch issue, which could lead to unreliable evaluation results [1]. Given that the dataset includes only about 200 instances, I suggest the authors clarify whether this issue might have caused noticeable variance in evaluation outcomes.
Additionally, providing a comparison on SWE-bench (e.g., Code Researcher vs. SWE-Agent) would make the evaluation more convincing. I understand that, as the authors note, smaller repositories may not fully leverage Code Researcher’s advantages. Still, even a limited-scale comparison would strengthen the credibility of the reported results. If the authors believe that these incomplete patches are unlikely to affect the results, a clear explanation of why this is the case would also suffice without extra experiments.

Reference
[1] https://github.com/Alex-Mathai-98/kGym-Kernel-Playground/issues/1

**Questions:**

1. What characteristics make Code Researcher “the first deep research agent for code”? How does its deep research capability fundamentally differ from that of standard software engineering or coding agents? If there is no essential difference at the task modeling level, I would suggest the authors narrow the claim and focus on the goal of solving large-scale and complex software development and maintenance tasks, rather than emphasizing the “deep research” concept.

2. How much does the memory filtering mechanism contribute to the overall performance? Could the authors provide ablation results or a qualitative analysis?

3. What unique mechanisms or algorithmic features enable Code Researcher to address kernel crash issues? It seems that most of its modules are consistent with those of regular agents (e.g., code search tools).

---

> ### Author Response · Authors · 2025-11-21
>
> Thank you for your insightful comments. We first address the dataset issue and then answer your questions.
>
> ## Dataset Issue
>
> We are aware of the linked issue, and would like to clarify that it does not affect our evaluation results at all. The issue points out that for some bugs (specifically, 5 out of 279 bugs), the ground-truth patch present in the dataset is incomplete (i.e., it does not have some of the changes that the human developer made). However, the validation of Code Researcher's patches is based on running the reproducer that triggered the crash and does not involve comparison with the ground-truth patch in the benchmark.
>
> ## Clarification on "Deep Research Agent for Code"
>
> We use "deep research" in the sense of [R1], involving extensive search, hypothesis generation, and pattern recognition across heterogeneous sources. Our task meets these criteria due to:
> 1. Large-scale search over the 28M-line codebase and 1M-commit history,
> 2. Hypothesis formation for crash causation and patch synthesis, and
> 3. Pattern recognition to detect coding conventions and violations.
>
> These elements are essential because the Linux kernel contains decades of evolving subsystems with subtle cross-file interactions. For example, memory allocation uses nonstandard kernel-specific APIs instead of standard C library calls, and discovering such conventions requires broad contextual exploration. Existing agents typically focus on a narrow window around the suspected edit site and thus fail to identify such global dependencies. The effectiveness of our deep-research strategy is reflected in Code Researcher’s stronger results compared to prior agents (58% crash resolution vs. 37.5% for SWE-agent) and its deeper codebase exploration (10 vs. 1.33 unique files).
>
> We also attempted to use ChatGPT Deep Research on the kernel repository; it reported that the repository was too large to index, underscoring the need for the scalable search tools we introduce.
>
> [R1] A Comprehensive Survey of Deep Research: Systems, Methodologies, and Applications, June 2025
>
> ## Importance of Memory Filtering
>
> Filtering’s contribution is demonstrated in "Context filtering" (Line 435, RQ4). The average context length drops from ~21.6K to ~7.8K tokens after filtering, which helps reduce irrelevant information. In an ablation over 20 crashes, disabling filtering reduced resolved crashes (10 to 8), recall (0.41 to 0.35), and All/Any/None metrics (34/16/50 to 29/15/56). This shows that combining broad context gathering with targeted filtering improves Code Researcher’s performance.
>
> ## Novel Features of Code Researcher
>
> Code Researcher differs from prior agents through its explicit context-gathering phase, its Analysis-Synthesis workflow, and several methodological innovations.
>
> ### Analysis-Synthesis Flow and Explicit Context Gathering
>
> The Analysis phase gathers context from diverse parts of the codebase before making any edits, reflecting how developers reason about large, interdependent systems. This workflow prevents premature editing and allows the agent to trace call chains, check invariants, and inspect related subsystems before proposing a fix. It aligns with workflows in Deep Research agents in other domains [1,2]. In contrast, SWE-agent follows a single edit-test loop that limits exploration, while Agentless focuses on locating edit points rather than gathering broader supporting context. These differences contribute to Code Researcher’s higher crash-resolution rate (48% vs. 31.5% for SWE-agent) and deeper exploration.
>
> [1] Gemini Deep Research (Overview page)
>
> [2] Researcher agent in Microsoft 365 Copilot (Microsoft 365 Copilot Blog, 28/3/2025)
>
> ### Concrete Novel Features
>
> 1. **Commit history analysis**: Although commit history is widely used in software engineering, no prior coding agent has integrated it systematically. We demonstrate its significance with a targeted ablation.
>
> 2. **Memory filtering**: For long-context agents, memory design is critical. Our filtering mechanism substantially reduces context size (from ~21.6K to ~7.8K tokens) and improves performance, as shown above and in RQ4.
>
> 3. **Scalable search tooling**: Prior tools (e.g., LocAgent [3], RepoGraph [4]) scale poorly even on mid-sized Python repositories. On the ~433K-LOC sympy repo, LocAgent required 764 seconds and RepoGraph errored after partial progress. Such approaches depend on dependency-graph construction, which grows rapidly with project size, making them unsuitable for repositories with tens of millions of lines of code. In contrast, constructing a `ctags` index for the same repo took 0.76 seconds, and for the 28M-LOC Linux kernel it takes only 74 seconds, enabling efficient large-scale exploration. See Appendix B, Line 747 for complete details.
>
> [3] Chen, Zhaoling, et al. "Locagent: Graph-guided llm agents for code localization." ACL 2025.
>
> [4] Ouyang, Siru, et al. "RepoGraph: Enhancing AI Software Engineering with Repository-level Code Graph." ICLR 2025.

---

> > ### Comment · Reviewer_SMcB · 2025-11-25
> >
> > Thanks for clarification. I am not very convinced by the statement of Code DeepResearch, as the following features can be achieved via adding corresponding prompt to existing general agents:
> > 1. "prevents premature editing and allows the agent to trace call chains": Generally speaking, it is nearly impossible for agent and even human developers to write a good code change without knowing its dependency relation. Therefore, fetching sufficient context is a common practice nowadays. For example, Mini-SWE-agent is asked to analyze the codebase by finding and reading relevant files first, rather than finding an exact place to edit.[1]
> > 2. "commit history analysis": Mini-SWE-agent can also achieve it via bash with Git if you write an additional prompt for it.
> >
> > It makes me feel the contribution a little bit incremental. If we can adjust a non-DeepResearch agent to a DeepResearch agent easily via changing prompts, then I think the problem of lacking a Code DeepResearcher is not really challenging and critical in this area. It is hard to find any essential difference between Code DeepResearch agent and existing code agent in their framework design.
> >
> > [1] https://github.com/SWE-agent/mini-swe-agent/blob/8c2d51bb72dc4fb56e58eb222d4e673bdd865b0e/src/minisweagent/config/mini.yaml#L27

---

> > > ### Author Response · Authors · 2025-11-26
> > >
> > > We respectfully disagree with the reviewer's conclusion that existing coding agents can be changed to Code Researcher with just a prompt change. The reviewer mentions Mini SWE-agent, which is a simpler version of SWE-agent. In the paper, we have compared against SWE-agent and shown that Code Researcher performs significantly better on crash resolution rate and on all other metrics. Code Researcher differs fundamentally from existing coding agents along the following axes:
> > >
> > > 1. **Different Phases vs. Single ReAct Loop**: Instead of a single ReAct loop, as in SWE-agent, Code Researcher shifts all the reasoning earlier in the Analysis phase (which is designed as a ReAct loop), which is then followed by Filtering and Synthesis (both one-shot LLM generations). This modular design addresses the unique challenges posed by large systems codebases, namely, that it is time and compute expensive to test changes, and that the context gathered in a ReAct loop quickly blows up. As a result, Code Researcher's behavior is quite different from existing agents, as we will explain in the next two points.
> > >
> > > 2. **Deeper Exploration vs. Trial and Error Refinement**: A repository like the Linux kernel takes around an hour to build, and another 10 minutes to try to reproduce the crash. There are indeed many such repositories, as foundational systems repositories often have long build times. However, the field of coding agents has largely ignored these repositories, and existing agents (like SWE-agent) get their performance on SWE-bench by repeatedly making changes and testing them in a loop. In contrast, Code Researcher has a focused exploration phase where it uses carefully designed reasoning strategies and search tools to gather all the relevant context from the repository. As we show, this results in much deeper exploration than prior agents.
> > >
> > > 3. **Scalable Search Tooling and Memory Filtering**: Because coding agents had been confined to moderately sized Python repositories, no one had paid attention to the design of scalable search tools. As we have shown in the paper, prior tools that try to use the control flow dependencies (e.g., LocAgent, RepoGraph) scale poorly even on mid-sized Python repositories. Similarly, the Filtering technique in Code Researcher addresses the challenge of context explosion that becomes critical in large systems codebases. This had also not been used in any prior coding agents.
> > >
> > > In particular, to address the points raised by the reviewer, we note the following.
> > >
> > > 1. You are right that it is common practice to add to the prompt to fetch sufficient context, however our results show that it is not nearly as effective as Code Researcher's approach. The prompt that SWE-agent uses (see `SWE-agent/config.yaml` in our supplementary material) has the line "Locate relevant code using the find and search commands.", similar to the line you have pointed out for Mini SWE-agent. Despite this, SWE-agent explores only 1.33 files per trajectory (as compared to 10 for Code Researcher) and resolves 16.5% fewer crashes than Code Researcher. This points to the importance of Code Researcher's explicit Analysis phase, equipped with reasoning strategies and scalable search tools.
> > >
> > > 2. In principle, it is possible for any coding agent that has `bash` access (as most agents do) to use `git` and perform commit search. However, no prior work has made use of the commit history. Code Researcher is the first coding agent to do so. It is also the first to demonstrate, with evidence, the need and utility of analyzing the commit history, especially for large systems codebases. So this contribution is a methodological one. Further, in our experience designing Code Researcher, even though SWE-agent can technically use `git` for commit search, it is not simple to elicit the behavior out of it that Code Researcher's design achieves.
> > >
> > > Taken together, the modular Analysis-Synthesis design, the reasoning strategies, and the scalable search tools of Code Researcher, help shift its behavior to intensive context gathering over large systems codebases, the main problem we address in our paper. We have carefully designed Code Researcher for this critical and challenging class of tasks, as shown by our results, where Code Researcher achieves 48% crash-resolution rate and explores 10 files on average per trajectory, as compared to just 31.5% and 1.33 files for SWE-agent.

---

### Official Review · Reviewer_Kutf · 2025-11-01

**Soundness:** 3
**Presentation:** 3
**Contribution:** 3
**Rating:** 6
**Confidence:** 4

**Summary:**

This paper proposes Code Researcher, a deep research-oriented agent for large-scale system code, aimed at fixing crash issues in complex codebases like the Linux kernel. The core innovation lies in introducing the deep research paradigm into the code repair domain, gathering necessary information through multi-step reasoning, historical commit analysis, and structured context memory. On the kBenchSyz benchmark, Code Researcher achieves a 58% crash resolution rate, significantly outperforming SWE-agent's 37.5%. Overall, this is a paper with solid engineering implementation and reasonable experimental design, but it has certain limitations in methodological innovation and evaluation comprehensiveness.

**Strengths:**

1. **Clear Problem Identification and Motivation**: It clearly identifies the shortcomings of existing code agents in handling large-scale system code (lack of deep context collection capabilities) and designs targeted solutions.

2. **Systematic Method Design**:
   - The three-stage process (analysis-synthesis-verification) is logically clear.
   - The reasoning strategies are reasonably designed (control/data flow tracing, pattern detection, commit history causal analysis).
   - The structured memory mechanism effectively manages complex contexts.

3. **Important Technical Contribution - Commit History Search**: It is the first to systematically integrate historical commit analysis into code repair agents, with ablation experiments proving its importance (performance drops by 10% when removed).

4. **Thorough Experimental Validation**:
   - Multi-dimensional comparative experiments (assisted vs. unassisted settings).
   - Detailed ablation studies.
   - Cross-codebase generalization validation (FFmpeg).
   - Qualitative analysis classification (accurate/over-specialized/incomplete/incorrect).

5. **Transparent Experimental Setup**: It provides details on data validation, computational resource usage, baseline adaptations, etc., enhancing reproducibility.

**Weaknesses:**

### Methodological Limitations
1. **Limited Innovation**: The core method essentially applies known techniques from deep research agents (multi-step tool calls, reasoning strategies, memory mechanisms) to the code domain, lacking fundamental innovations tailored to code characteristics.

2. **Generality of Reasoning Strategies**:
   - The three reasoning strategies (control/data flow, pattern detection, commit history) are reasonable but are direct applications of traditional software engineering concepts.
   - It fails to fully leverage code's structured characteristics (e.g., AST, program dependence graphs).
   - It lacks specialized reasoning for system code-specific complexities (concurrency, memory management, hardware interactions).

3. **Simplistic Context Filtering Mechanism**: The synthesis stage relies solely on LLM to judge memory relevance, lacking more refined filtering strategies based on program analysis.

### Evaluation Deficiencies
4. **Representativeness Issues in Benchmark Dataset**:
   - kBenchSyz only includes fuzzing-discovered crashes, potentially biased toward specific bug patterns.
   - Out of 279 instances, only 200 (71.7%) can be reproduced, raising questions about data quality.
   - It lacks analysis of the distribution across different bug types (memory errors, concurrency, logic errors).

5. **Single-Metric Patch Quality Evaluation**:
   - Using only "whether it prevents the crash" as the success criterion is too coarse.
   - While qualitative analysis includes classifications, it lacks systematic quantification (what are the proportions of each category in the 58% patches?).
   - It does not evaluate patch side effects (performance impact, functionality disruption, maintainability).

6. **Unfair Comparison with CrashFixer**:
   - CrashFixer is evaluated in the assisted setting (known buggy files).
   - Budget settings are incomparable (P@16 vs. P@5).
   - The authors admit "exact budget is not known," reducing the persuasiveness of the comparison.

7. **Missing Important Baselines**:
   - No comparison with AutoCodeRover (despite mentioning it in the paper).
   - No testing of simple long-context baselines (putting entire crash-related files into context).
   - No comparison with RAG-based methods.

8. **Insufficient Generalization Validation**:
   - FFmpeg experiment uses only 10 samples, lacking statistical significance.
   - No testing on other system codebases (e.g., database systems, network stacks).
   - No testing on non-C/C++ language system code.

### Technical Detail Issues
9. **Search Efficiency and Scalability**:
   - Average exploration of 29 files per bug; scalability on larger codebases (e.g., Chromium) is unknown.
   - Performance bottlenecks of search tools (ctags, git grep) are not discussed.
   - The basis for the 60-second timeout setting is unclear.

10. **Limitations of Commit History Search**:
    - Supports only regex search, unable to handle semantically similar but differently phrased commits.
    - Does not consider commit recency (recent commits may be more relevant).
    - Truncating to 100 lines may lose important context.

11. **Temperature Parameters and Sampling Strategies**:
    - Analysis uses 0.6 temperature, synthesis uses increasing temperatures (0, 0.3, 0.6), lacking ablation experiments for validation.
    - Why not try diversified sampling in the analysis stage as well?

**Questions:**

1. **Sufficiency of Reasoning Strategies**: The paper proposes three reasoning strategies, but how do they ensure coverage of the main diagnostic patterns for system code crashes? Have strategies based on program analysis tools (e.g., static analysis, symbolic execution) been considered?

2. **Optimization of Context Selection**: In RQ3, Code Researcher reads 29.1 vs. 1.9 files compared to SWE-agent, but is "more" always "better"? Are there counterexamples where excessive irrelevant context reduces performance? **Moreover, as far as I know, Gold Patches on SWE-Bench often require modifying only 1-3 files.**

3. **Necessity Analysis of Commit History**:
   - Among the 96 successful cases, how many truly rely on commit history?
   - Can examples be provided to illustrate which types of bugs must depend on historical commits to fix?
   - Is the method still effective for projects with shorter codebase histories?

4. **Comparison with Human Developers**:
   - How many files do humans typically explore to fix these bugs?
   - How similar is Code Researcher's reasoning path to human experts' diagnostic processes?
   - Are there cases where Code Researcher discovered contexts unnoticed by humans?

5. **Error Propagation and Cascading Fixes**:
   - How to handle situations where a crash requires modifying multiple independent modules?
   - How does the synthesis stage determine the scope of modifications?
   - Are there cases where partial fixes lead to new issues?

6. **Cross-Version Generalization**:
   - Experiments are conducted on parent commits; what if tested on earlier versions (e.g., parent's parent)?
   - Can the method migrate knowledge across different kernel versions?

7. **Industrial Applicability**:
   - What is the average reasoning time per bug?
   - Is a 58% resolution rate sufficient for practical value in actual development workflows?
   - How to integrate with existing CI/CD processes?

8. **Potential of Test-Time Scaling Methods**:
   The paper only tests simple test-time scaling in the Unassisted + Scaled setting (increasing max_calls and sampling number k), finding limited effects from increasing trajectory length but some help from increasing sampling diversity. However, recent advanced TTS methods on SWE-Bench, such as SE-Agent [1]'s self-evolution trajectory optimization, can significantly improve performance through iterative refinement and trajectory quality assessment. Can these advanced TTS methods be migrated to this approach and bring substantial improvements? This also affects the method's generalizability.

**Among these, questions 1, 2, and 8 are more important; if my concerns can be addressed, I am inclined to raise the score.**

---
References: [1] SE-Agent: Self-Evolution Trajectory Optimization in Multi-Step Reasoning with LLM-Based Agents

---

> ### Author Response · Authors · 2025-11-21
>
> Thank you for the insightful comments and for recognising the importance of the commit history search contribution. We answer key questions first and then address the other questions and weaknesses.
>
> ## Answers to Questions
>
> 1.  **Sufficiency of Reasoning Strategies**: Thank you for the question. Our strategies (control flow, pattern search, commit analysis) are based on domain knowledge of the complexities of large codebases. Given the crash stack trace, it is natural to follow the control flow and understand how data is passed between functions (Strategy 1). Developers also search for patterns, especially if they find a suspicious code snippet and want to check if it is common (Strategy 2). Commit history analysis (Strategy 3) helps them identify which change introduced the bug and why it was made. Code Researcher resolves **58%** (with GPT-4o+o1) and **67%** (with Gemini 2.5-Flash) of crashes, showing these strategies are effective.
>
>     We considered program analysis tools (LocAgent [1], RepoGraph [2]) but found they scale very poorly. For the $\sim 433\text{K}$ LoC sympy repository: LocAgent took **764 seconds**, RepoGraph errored out after **44.3 seconds** with an estimated time of **30 minutes**. In contrast, `ctags` took only **0.76 seconds**. Scaling dependency graphs to the Linux kernel ($\sim 28\text{M}$ LoC) is infeasible, while `ctags` takes only **74 seconds**.
>
>     [1] Chen, Zhaoling, et al. "Locagent: Graph-guided llm agents for code localization." ACL 2025.
>
>     [2] Ouyang, Siru, et al. "RepoGraph: Enhancing AI Software Engineering with Repository-level Code Graph." ICLR 2025.
>
> 2.  **Optimisation of Context Selection**: You are correct that quality matters. In RQ3, we show that Code Researcher gathers not only more context, but also higher quality context:
>
>     i) **Overlap with developer-referenced context**: Our context has a **63.7% overlap** (vs. **54.18%** for SWE-agent) with context developers reference in fix commit messages. Crucially, **30.8%** of our patches share overlapping commit context with the developer's analysis.
>
>     ii) **Context quality when both edit all the ground-truth modified files**: When both agents edit the correct files, Code Researcher's success rate is **61.1%** while SWE-agent's is **37.78%**. Code Researcher is the first agent to explicitly gather additional global context and this shows that it makes a huge difference.
>
>     Finally, note that the memory filtering mechanism before Synthesis is also designed to filter out irrelevant context. It reduces context length from $\sim 21.6\text{K}$ to $\sim 7.8\text{K}$ tokens, and omitting it drops performance by **10%** (RQ4, Line 435).
>
> 3.  **Necessity Analysis of Commit History**: We agree that it is important to analyse how useful commits are. An ablation study (Line 427, RQ4) showed removing the `search_commits` action leads to a **10% drop** in the crash resolution rate and decreases the ability to edit ground-truth files. This highlights its crucial role in context gathering. The method is also effective for projects with shorter histories as the relevant commits would be easier to find.
>
> 4.  **Comparison with Human Developers**: Code Researcher is designed to gather the **additional context** that human developers hold in mind when making edits in complex codebases, going beyond localising the edit. Our results on context overlap (see answer to Q2 above) show it achieves this well.
>
> 5. **Error Propagation and Cascading Fixes**: Indeed, a crash may involve multiple independent subsystems, and because all paths start from a common entrypoint, this is almost certain. The control and data flow reasoning strategy helps Code Researcher handle this issue and explore all the involved subsystems. This results in the Analysis phase collecting all the relevant context from multiple kernel modules. The Synthesis phase then analyses this context and proposes fixes, which may edit one or more kernel subsystems. Patches can be Inaccurate (fixing the crash but differing from the human's solution); for example, in Appendix J (Example D), Code Researcher adds defensive checks rejecting certain port numbers rather than handling them, as done by the developer patch.
>
> 8\. **Potential of Test-Time Scaling Methods**: Thank you for the suggestion, this is indeed an intriguing direction to investigate. The self-evolution trajectory optimisation approach studied by SE-agent is useful because its exploration is guided by previous trajectories and it can combine the good aspects of multiple trajectories into a better overall reasoning strategy. This approach is certainly complementary to Code Researcher's Analysis phase. By reflecting on multiple such trajectories, it can find and explore a better reasoning trajectory. Our results in the paper already show that simple test-time scaling by sampling more trajectories helps, so more powerful methods such as this should perform even better given more time and compute.

---

> ### Author Response · Authors · 2025-11-21
>
> 6\. **Cross-Version Generalisation**: We run Code Researcher on the parent of the fix commit for consistency with the benchmark setup, but it can be run on any buggy commit. The dataset contains crashes across multiple major and minor kernel versions, and the `search_commits` functionality allows Code Researcher to fetch knowledge from previous kernel versions.
>
> 7\. **Industrial Applicability**:
>
> (i) Deep Research agents are typically given time in the order of minutes to hours. While our LLM API access was rate-limited, our tooling is **scalable** to the Linux kernel with more than **28 Million Lines of Code** (see answer to Q1 above).
>
> (ii) Given the complexity of the Linux kernel, a **58%** resolution rate (**67%** with Gemini 2.5-Flash) is sufficient for practical value. Code Researcher is valuable because it gathers relevant context and generates a hypothesis about the root cause, enhancing interpretability for developers.
>
> (iii) Code Researcher could be run as part of the Linux kernel's existing CI/CD fuzzing pipeline to produce candidate fixes.
>
> ## Addressing the Weaknesses
>
> ### Innovation of Code Researcher (1)
>
> We believe that our innovations are more than incremental. What fundamentally differentiates Code Researcher from other coding agents is the Analysis phase which iteratively gathers context from different parts of the codebase *before* making edits. This Analysis-Synthesis workflow is also consistent with how Deep Research agents in web or document domains operate. Code Researcher differs from SWE-agent, which follows a single iterative flow of making edits and testing, and from Agentless, which follows a fixed workflow to identify the locations to *edit*, not the additional context that is important for making the edit.
>
> Code Researcher also has three concrete novel innovations that are not found in any prior work, namely, Commit History Analysis, Memory Filtering, and Scalable Search Tooling for a 28 Million LoC codebase. For a more detailed discussion, please refer to our explanation of "Novelty of Code Researcher" in our response to Reviewer YXzc.
>
> ### Strategies Based on Program Analysis (2)
>
> We have considered strategies based on program analysis tools. We have performed an experimental evaluation of prior coding agents using such tools and shown that they scale very poorly even to large Python codebases, making it extremely hard to scale them to codebases like the Linux kernel within a reasonable time/cost budget. Please refer to our answer to Question 1 above.
>
> ### Effectiveness of Filtering Mechanism (3)
>
> Filtering’s contribution is demonstrated in "Context filtering" (RQ4). The average context length drops from ~21.6K to ~7.8K tokens, helping reduce irrelevant information. In an ablation on 20 crashes, disabling filtering reduced resolved crashes (10 to 8), recall (0.41 to 0.35), and All/Any/None metrics (34/16/50 to 29/15/56). This shows that combining broad context gathering with targeted filtering improves Code Researcher’s performance.
>
> ### Diversity of Bugs in Benchmark (4)
>
> The kBenchSyz benchmark is canonical for a large, complex systems codebase ($28\text{M}$+ LoC, $75\text{K}$+ files, $1\text{M}$+ commits). It includes a healthy mix of bug patterns (memory leaks, concurrency, use-after-free, double-free) across versions and subsystems. For example:
>
> * Use-after-free bug: ID 362b5b49fced29361c0ba3007a2b3e5cd13776b5.
> * Concurrency bug/deadlock: ID f080ff8eb73ae6a29960594307de8e732db389d3.
> * Largest crash report (~56K lines): ID 1860a98f3556d97065ad773d095aa1d7eb5fbafa.
>
> ### Patch Evaluation (5)
>
> Beyond the standard kBenchSyz procedure, we also run kernel unit tests when available, thus our evaluation is not single metric. Despite the wide range of kernel versions in our dataset, we validate 88 of 116 patches with unit tests, and none violate existing tests (RQ5). We additionally provide qualitative analyses and complete examples of patch categories in Appendix J.
>
> Kernel crash reproduction is challenging due to nondeterminism and concurrency. We test patches in isolated VMs running the original reproducer, using four VMs in parallel and with multiple concurrent processes per VM. A crash in any VM within 10 minutes is counted as a failure. Full details are in Appendix G.
>
> ### Comparison with CrashFixer (6)
>
> We respectfully disagree that our comparison with CrashFixer is unfair. First, the code is not open-sourced, so we could not run any experiments and could only present the numbers mentioned in their paper. Second, the evaluation in fact gives an **unfair advantage to CrashFixer**. This is because:
>
> (i) CrashFixer is evaluated in the assisted setting where the buggy files are already given while Code Researcher is just given the crash report.
>
> (ii) CrashFixer uses a higher sampling budget of Pass@16, as compared to the Pass@5 budget used by Code Researcher.
>
> Despite this unfair advantage to CrashFixer, Code Researcher still beats/is competitive with its numbers.

---

> > ### Author Response · Authors · 2025-11-21
> >
> > ### Baselines (7)
> >
> > (i) Many different coding agents have been developed for SWE-bench. We have chosen SWE-agent because it is a state-of-the-art open-source agent. In particular, SWE-agent is also above AutoCodeRover on the SWE-bench Verified leaderboard.
> >
> > (ii) We have tested long-context baselines too, including both GPT-4o and o1 in the stack context setting. Code Researcher beats o1 and GPT-4o in the stack context setting, resolving 18% and 8.5% more crashes respectively. Full results are in Table 1 in RQ1.
> >
> > (iii) We also evaluated the feasibility of using RAG-based methods. Through an experiment, we estimated roughly 108 hours of embedding time for a single kernel snapshot, making this infeasible in our setting. Full details are in "Rationale for omitting embedding-based retrieval" (Appendix G, Line 1285).
> >
> > ### Search Efficiency and Scalability (9)
> >
> > Through experimental comparison against prior coding agents with other kinds of search tools, we have shown that their search techniques scale poorly even to large Python codebases, making it infeasible to use them within a reasonable time/compute budget for a codebase like the Linux kernel. Please refer to our answer to Question 1 above and to "Scalability of search tooling" (Appendix B, Line 747) for a detailed comparison of our search tooling with prior work.
> >
> > ### Limitations of Commit History Search (10)
> >
> > Our commit history search does take into account commit recency as it returns the most recent commits matching the regex first. We will point this out in the paper.

---

### Official Review · Reviewer_YXzc · 2025-11-11

**Soundness:** 3
**Presentation:** 3
**Contribution:** 2
**Rating:** 4
**Confidence:** 4

**Summary:**

This paper introduces Code Researcher, a deep research agent for code. It first performs an analysis phase to gather context, then a synthesis phase to filter irrelevant context and generate a patch, and finally a validation phase to test the patch. Empirical evaluation is provided on fixing system code crashes (kBenchSyz). Additional analyses are done to evaluate Code Researcher's context gathering accuracy and some ablations on its design.

**Strengths:**

1. This paper focuses on fixing system code bugs, which is related to the popular SWE-bench type problems but more low level and potentially of a larger scale. Given the importance of system code, developing more effective systems for that is meaningful. Empirical evidences suggest Code Researcher outperforms existing agents for general software, such as SWE-agent and Agentless.

2. This paper contains many insightful analyses on the Code Researcher system, such as the accuracy of the Analysis phase and inference scaling impact on the resolution rate. They help a reader better understand the system.

3. Overall the paper is well-written and easy to follow.

**Weaknesses:**

1. The novelty for Code Researcher is limited. It follows a three-phase workflow similar to Agentless. Its Analysis phase is iterative. Likewise, SWE-agent and Openhands can also perform iterative context retrieval. What makes Code Researcher different seems to be: 1) additional tools (search_commits); 2) a specialized search strategy guidance through prompting (section 3.1.2). So the overall innovations are incremental.

2. The Pass@k metric seems incomplete as described in the paper. It is defined as "prevents the crash" (line 273). This does not mean the patch is functionally correct. Section 5.5 provides some additional tests but they are incomplete as well, for example 28 out of the 116 crashes analyzed do not have additional tests. The ambiguity in correctness measure makes it difficult to evaluate the reported numbers. A clear correctness measure is necessary to evaluate coding agent systems.

**Questions:**

1. What is the search action distribution for the actions mentioned in section 3.1.1?

2. In the experiments, $k$ trajectories are sampled in the Analysis phase. Does the Synthesis phase use the combined context from all trajectories or does it use each one separately?

3. Why is 50k token chosen as the context length limit? Both GPT-4o and o1 can handle much longer context.

4. I find it interesting that increasing the number of trajectories is more effective than increasing their lengths. Is there an explanation why this is the case?

5. In addition to measuring the ground-truth file recall on patches, it would also be valuable to evaluate on the retrieved context.

---

> ### Author Response · Authors · 2025-11-21
>
> Thank you for the thoughtful comments and for highlighting systems code as an important domain. We address your questions first, then summarise the novelty and evaluation.
>
> ## Answers to Questions
>
> 1. **Search Actions Distribution**:
>    We analysed trajectories from the GPT-4o, P@5 run. Mean/median counts per trajectory were `7.02/7` `search_definition`, `6.89/6` `search_code`, and `4.66/4` `search_commits`. 94.97% of trajectories used all three actions, showing that Code Researcher consistently uses all its actions during Analysis.
>
> 2. **Trajectory Sampling Process**:
>    Each trajectory independently performs an Analysis, Synthesis, and Validation phase. Thus, sampling $k$ trajectories produces $k$ independently gathered contexts and $k$ synthesised patches, one from each context.
>
> 3. **Choice of the 50K Context Limit**:
>    The 50K limit was partly imposed by the APIs available to us. In practice it was exceeded in only 55 of 1000 trajectories. Code Researcher stays token-efficient because (i) patch generation only reads the gathered context rather than the full trajectory, (ii) this context is filtered to what is relevant (see “Context filtering”, RQ4), and (iii) it uses targeted `search_definition` and `search_code` calls rather than `read_file` actions used in some prior agents.
>
> 4. **More Trajectories vs. Longer Trajectories**:
>    Our examination of failures showed that when a trajectory adopts an incorrect hypothesis, it rarely backtracks. Independent trajectories are more effective because they can explore different lines of thought. We will add this discussion, thank you for the suggestion.
>
> 5. **Evaluating Retrieved Context**:
>    Beyond Recall (which checks whether the right files are edited), we evaluate whether the *additional* context Code Researcher gathers aligns with what developers relied upon. As shown in RQ3, its gathered context overlaps with developer-referenced commit content by 63.7% (vs. 54.18% for SWE-agent), and 30.8% of its produced patches overlap with developers’ own commit-level context. Furthermore, when both systems correctly edit all ground-truth files, Code Researcher succeeds in 61.1% of cases vs. SWE-agent’s 37.78%, highlighting the value of the retrieved context.
>
> ## Novelty of Code Researcher
>
> ### Analysis–Synthesis Reasoning and Explicit Context Gathering
>
> Code Researcher differs fundamentally from prior coding agents through its explicit Analysis phase that gathers global context *before* any edits are proposed. This reflects how human developers approach large, interdependent codebases and is consistent with emerging “deep research”-style agent workflows in other domains. In contrast, existing SWE agents run a single iterative edit-test-reflect loop without first exploring the broader codebase. This results in far shallower exploration (e.g., SWE-agent averages 1.33 files per trajectory vs. Code Researcher's 10) and weaker performance (48% crash resolution vs. 31.5%). Agentless focuses on edit localisation but does not collect the broader context (see also Q5 above) needed for high-quality patches.
>
> ### Concrete Novel Contributions
>
> 1. **Commit History Analysis**:
>    Code Researcher is the first system to make systematic use of historical commits in code-repair agents. Ablations show that commit-level insight is often essential for identifying root causes and is missing from prior tools.
>
> 2. **Memory Filtering**:
>    Before Synthesis, Code Researcher filters the gathered context to keep only the most relevant parts. As shown in RQ4, this reduces the average context size substantially and its removal leads to a 10% performance drop on a sample subset, confirming its importance.
>
> 3. **Scalable Search Tooling**:
>    We experimentally compare commonly proposed code-graph and dependency-analysis tools to the lightweight `ctags`-based index used by Code Researcher (Appendix B, Line 747). Systems like LocAgent and RepoGraph scale poorly even on ~433K-LOC Python codebases: LocAgent took 764s; RepoGraph failed after 44.3s while still projecting >30 minutes to finish. In contrast, Code Researcher’s `ctags` index requires no dependency analysis and finishes in 0.76s on the same repo. Given that the Linux kernel has ~28M LOC, such dependency-graph approaches are infeasible, whereas `ctags` indexing completes in 74 seconds.
>
> ## Patch Evaluation
>
> Kernel crash reproduction is challenging due to nondeterminism and concurrency. We test patches in isolated VMs running the original reproducer, using four VMs in parallel and with multiple concurrent processes per VM. A crash in any VM within 10 minutes is counted as a failure. Full details are in Appendix G.
>
> Beyond the standard kBenchSyz procedure, we also run kernel unit tests when available. Despite the wide range of kernel versions in our dataset, we validate 88 of 116 patches with unit tests, and none violate existing tests (see RQ5). We additionally provide qualitative analyses and complete examples of patch categories in Appendix J.

---

### Meta-Review · Area_Chair_Hawy · 2026-01-05

**Summary:**

This paper proposes a code agent designed to enhance performance on Linux kernel crash benchmarks. Reviewers unanimously agree that the research direction is important and practically valuable: addressing crash repair for complex, large-scale system code (such as the Linux kernel) through an LLM-driven agent. The paper demonstrates solid engineering implementation, reasonable experimental design, and exhibits superior performance over existing general-purpose coding agents (e.g., SWE-Agent) on the kBenchSyz benchmark.

Reviewers acknowledge that the problem selection is challenging and practical, the system design is systematic and clear, and the work clearly identifies the limitations of existing code agents in critical system kernel research, conducting an in-depth investigation into a more demanding problem. They also recognize the effective experiments and insightful analysis presented in the paper.

Reviewers have raised the following primary concerns:

1. The reviewer concerns about innovation, the methodological innovation of the agent compared to existing agents is incremental, primarily through the addition of a search_commits tool. Doubts remain about whether the use of commit history would be effective on other benchmarks or tasks.

2. Limited test samples: Reviewers are concerned about the agent's generalization capability due to the small number of test cases.

3. Overly coarse evaluation criteria for patch quality: The primary success criterion is "whether the crash is prevented," which does not guarantee functional correctness or the absence of side effects (such as performance impact or disruption of other functionalities). While qualitative analysis includes categorization, it lacks quantitative statistics on the proportion of each category.

**Reviewer Concerns:**

In response, the authors have addressed some of the reviewers' concerns. Although they explained the reasons for not testing on SWE-Bench, no concrete actions were taken to alleviate concerns about generalization capability.

**Reviewer Scores:**

After thorough discussion, reviewers are inclined to maintain their current scores.

---

### Decision · Program_Chairs · 2026-01-26

Reject